# TreeMoCo: Contrastive Neuron Morphology Representation Learning

**Hanbo Chen**[*,†]
Tencent AI Lab
hanbochen@tencent.com

**Jiawei Yang**[*,†]
University of California, Los Angeles
jiawei118@ucla.edu

**Daniel Maxim Iascone**
University of Pennsylvania, Philadelphia
daniel.iascone@pennmedicine.upenn.edu

**Lijuan Liu**
Southeast University, Nanjing
juan-liu@seu.edu.cn

**Lei He**
University of California, Los Angeles
lhe@ee.ucla.edu

**Hanchuan Peng**
Southeast University, Nanjing
h@braintell.org

**Jianhua Yao**[‡]
Tencent AI Lab
jianhuayao@tencent.com

## Abstract

Morphology of neuron trees is a key indicator to delineate neuronal cell-types, analyze brain development process, and evaluate pathological changes in neurological diseases. Traditional analysis mostly relies on heuristic features and visual inspections. A quantitative, informative, and comprehensive representation of neuron morphology is largely absent but desired. To fill this gap, in this work, we adopt a Tree-LSTM network to encode neuron morphology and introduce a self-supervised learning framework named TreeMoCo to learn features without the need for labels. We test TreeMoCo on 2403 high-quality 3D neuron reconstructions of mouse brains from three different public resources. Our results show that TreeMoCo is effective in both classifying major brain cell-types and identifying sub-types. To our best knowledge, TreeMoCo is the very first to explore learning the representation of neuron tree morphology with contrastive learning. It has a great potential to shed new light on quantitative neuron morphology analysis. Code is available at https://github.com/TencentAILabHealthcare/NeuronRepresentation.

## 1 Introduction

The central nervous system is composed of hundreds to thousands of neuronal subtypes. Subtype is the critical determinant of neuronal function within local and long-range circuits and is identified by a combination of gene expression patterns, dendrite/axon morphology, synaptic connectivity, and electrophysiological properties. In particular, dendritic morphology is a central node of neuronal subtype identity, as dendritic branches receive the vast majority of the presynaptic input onto each neuron and directly shape the electrophysiological properties of the neuron that governs its firing output. Delineating neuron cell-types based on their dendritic morphology is one of the core problems of neuroscience for the past century [51].

---

[*]: Equal contribution. [†]: Work done when authors were at Tencent AI Lab. [‡]: Corresponding author.

36th Conference on Neural Information Processing Systems (NeurIPS 2022).

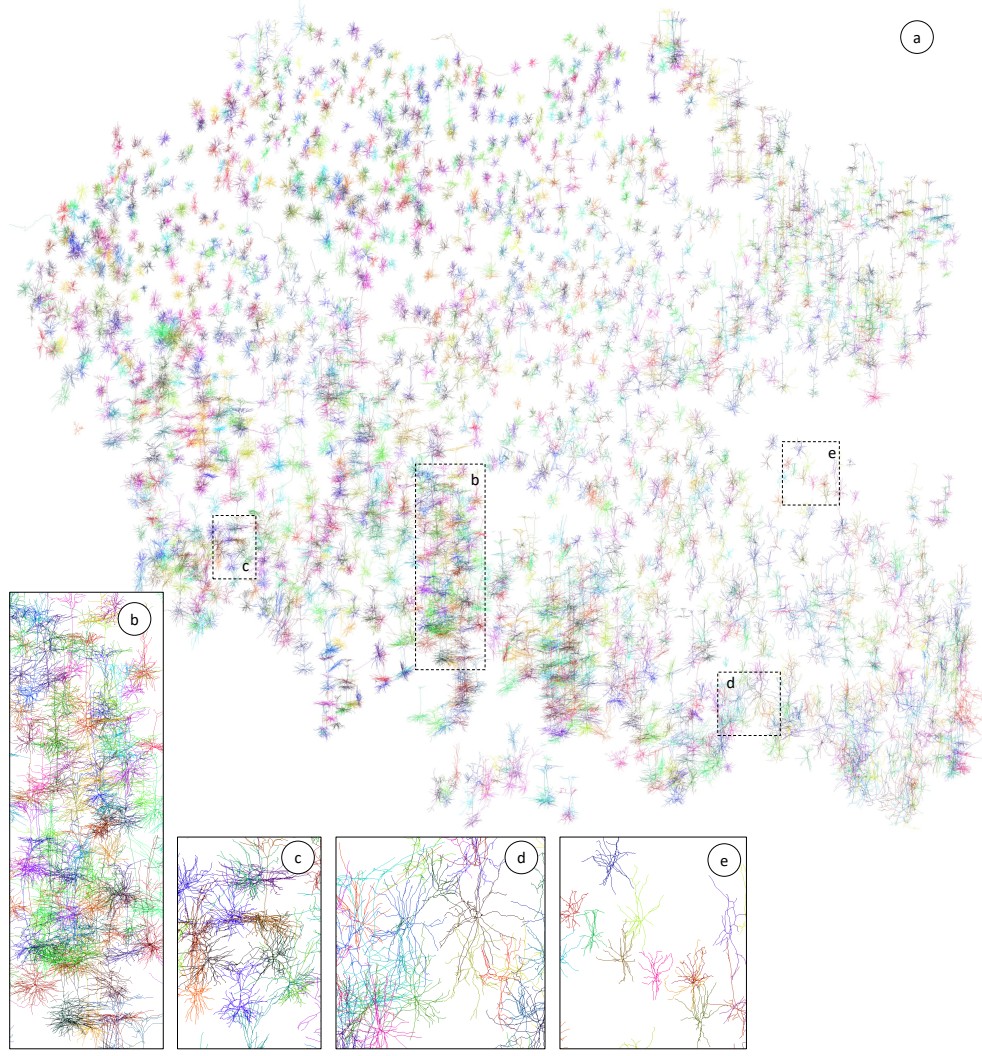

Figure 1: Distribution of neurons in the embedding space of TreeMoCo visualized by t-SNE [39]. Thumbnails of dendrites are randomly colored. (b-e) Zoom-in view of regions in (a) showing clustering of isocortex layer 5 cells (b), layer 2/3 cells (c), basket cells (d), and Martinotti cells (e).

Neuron dendritic morphology can be digitally represented by a 3D neuron reconstruction (Fig. 2(a)), a tree graph with 3D coordinates of each node; the root is placed at soma. With recent advances in imaging [9, 11, 47] and computational methods [20, 27, 42], 3D reconstruction of whole neuron trees with high-throughput is now accessible for mouse brain [28, 44]. However, due to the lack of effective measurement, most analyses of these 3D reconstructions are still largely based on biologists' visual inspection. Considering the enormous number of neurons in the brain and the 3D formation of neuron morphology, such visual inspection is subjective, biased, and has low throughput.

The past decade has witnessed the success of deep learning in learning representations of almost all kinds of data such as image, language, sound, and graph [2]. However, as a special type of graph or graphic data, most of the quantitative analysis of neuron trees is still based on simple heuristic features [33]. Only a few attempts have been made in learning deep representations of neuron morphology. In [52], the authors proposed to encode neuron morphology with tree recurrent neural network (TRNN). However, this method relies on manual annotations, limiting its application to supervised training only. In [22], the authors proposed a generative self-supervised learning (SSL) method for neuron morphology representation learning, named MorphVAE, to reconstruct original neuron trees via a variational autoencoder and random walk sampling. Since this method

cannot encode tree topology structure, its classification performance is limited and the method still relies on label information to generate reasonable predictions. In [32], the authors adopt multi-view based SSL to learn representations of neuron morphology. Though the work mainly focuses on the local components of neuron branch which is not available in whole neuron reconstruction data, its promising results advocate the possibility of learning morphology representation without labels.

Inspired by the contrastive learning successes in visual representation, we experiment with adapting MoCo for neuron morphology representation learning, namely TreeMoCo. It can learn embedding of 3D neuron reconstructions without needing prior knowledge such as cell type annotation. It first maps neuron trees input to a deep embedding space by our proposed neuron encoder, and then incorporates an SSL pipeline composed of neuron tree augmentations and a contrastive loss to learn representations. By testing the framework on 2403 neuron reconstructions from 3 different data resources, we are excited to find that TreeMoCo can clearly delineate neuron morphology in the embedding space (Fig. 1). Comparison with prior arts shows that TreeMoCo can achieve premium results than supervised methods even *without annotations*.

## 2 Background and Related Works

**Heuristic features of neuron morphology.** To quantitatively describe the neuron morphology, a set of heuristic features and related tools namely L-measure [33] has been proposed and widely used. These features can be summarized as 4 categories including (1) distance-based measurements (*e.g.*, branch length, distance to soma); (2) angle-based measurements (*e.g.*, bifurcation angle); (3) topology-based measurements (*e.g.*, number of bifurcations, branch order); and (4) size-based measurements (*e.g.*, branch radius, surface area). Statistics of these features such as mean or standard deviation are usually adopted to describe neuron morphology. Such representation is easy to compute and explain. However, they cannot differentiate complex morphologies (examples in Appx. A.1).

**Cell-type and neuron morphology classification.** As the primary function of dendritic branches is to compartmentalize functionally related synaptic inputs, dendritic morphology can be used to categorize neuronal subtypes based on total dendritic length, amount of branching across the dendritic trees, and localization of dendritic branch fields to different brain regions (such as layers of neocortex). Most existing cell-type classification algorithms rely on hand-crafted features [33] to predict or cluster cell types [40, 24]. In some recent works, researchers started to adopt tree-based model [52, 23] and sequence-based model [22] to classify neuron morphology. Despite competing performance than classic methods, training of these models requires manual annotations of cell types.

**Contrastive learning.** Contrastive learning (CL) is a SSL method that aims at bringing positive sample pairs closer and spreading away the negative sample pairs in the feature embedding space. This method dates back to [13] or earlier, and has recently drawn significant interest across different communities [45, 25, 17, 48, 16, 6, 5]. Some research efforts have been made in graph contrastive learning (GCL) [37, 30, 15, 49, 46]. Particularly, GraphCL [49] studies the graph data augmentations systematically for GCL, and InfoGCL [46] provides a theoretical analysis for it. However, to our best knowledge, no work has studied the tree graph, a special instance of graph, within the context of CL.

## 3 Methods

### 3.1 Tree graph representation of neuron

We define a neuron reconstruction as $T \triangleq (\mathbf{V}, \mathbf{E})$, where $\mathbf{V} = \{v_k\}; v_k = (x_k, y_k, z_k)$ are nodes with 3D coordinates and $E = \{e_k\}; e_k = (v_k, v_k^{parent})$ are edges connecting each node with its parent. As shown in Figure 2-(e), the number of 1-degree nodes could vary by reconstruction and image resolutions, while the bifurcation/tip nodes preserve major topology information. For computational efficiency, we remove 1-degree nodes and directly connect the root node, the bifurcation nodes, and the tip nodes in $T$ (Fig. 2(b)). To preserve the shape information of 1-degree node sequences (branch in Fig. 2(e)), we calculate the related features and concatenate them with the coordinate property of its downstream node, as illustrated in Figure 2-(e,f). The processed tree graph is defined as $\Upsilon \triangleq \mathrm{Process}(T) \triangleq (\mathbf{X}, \mathcal{G})$, where $\mathcal{G}$ represents the tree topology and $\mathbf{X} = \{\mathbf{x}_k\}_{k=1}^N$ is an $N \times (3 + Q)$ matrix that is from $N$ nodes with 3D coordinates and $Q$-dim *branch* features.

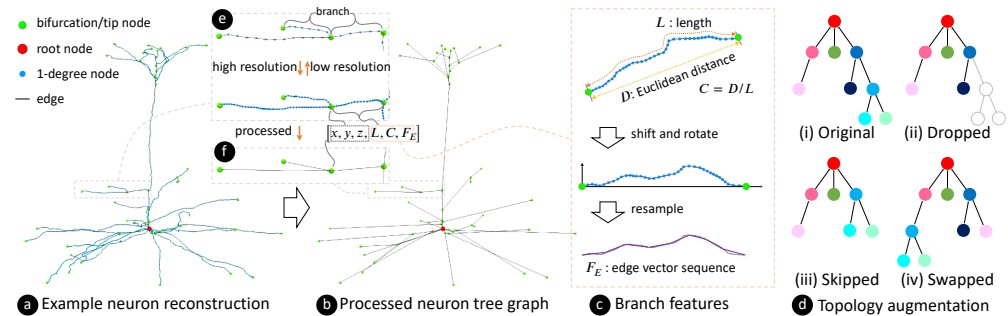

Figure 2: Illustration of (a) neuron reconstruction, (b) processed tree graph, (c) branch feature computation, (d) topology augmentations. (ii) to (iv) correspond to (9) to (11) in Table 1. (e) and (f) illustrate neuron tree processing for computation efficiency and resolution normalization purposes.

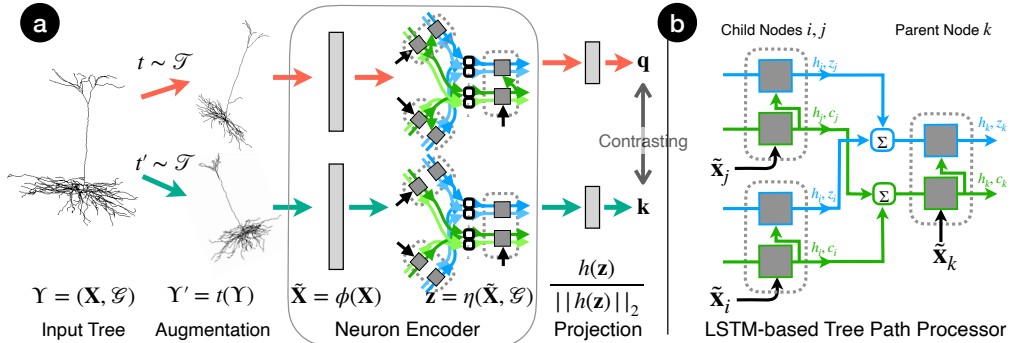

Figure 3: The overview of TreeMoCo: (a) TreeMoCo pipeline. (b) The zoom-in view of how the LSTM-based tree path processor $\eta$ propagates information from child nodes $i$ and $j$ to a parent node $k$. Blue and green colors denote the corresponding information flows for two LSTM cells, respectively. The symbol "$\Sigma$" denotes summation. The dashed rectangles mark our double-cell LSTM processor.

*Branch* feature is a $Q$-dim vector, of which two dimensions denote length and contraction. Length $L$ is the accumulated distance between adjacent nodes along a branch: $L = \sum_i^{start \leq i < end} ||v_i - v_{i+1}||_2$, which measures size. Contraction is the ratio between the Euclidean distance $D = ||v_{start} - v_{end}||_2$ and the length $L$ of a branch: $C = D/L$, which measures the curviness. Notably, both measurements are invariant to the order of edges and thus cannot differentiate certain patterns (details in Appx. A.1). To allow a comprehensive encoding of branch pattern, we resample each branch with a fixed number of vertices (8 in this paper) and take the sequence of the edge vectors $\vec{E_i} = v_i - v_{i+1}$ as the its shape feature: $F_E \in \mathbb{R}^{24}$. In addition, the branch orientation is unified before extracting feature vectors such that the start and end points are on the x-axis and the principle direction is aligned with y-axis (Fig. 2(c)). Together, there are 29 features for each neuron node: $\mathbf{x} = [x, y, z; L; C; F_E]$.

### 3.2 Neuron Encoder

We next propose a neuron encoder $f$ to project the neuron tree graph into a feature space. Our model has two components: a trainable MLP (multi-layer perceptron) embedding layer ($\phi$) that maps point coordinates and other node-related features into the embedding space, and a LSTM-based tree-path processor ($\eta$) that exploits tree topology information. Formally, we define the neuron encoder as $f \triangleq \eta \circ \phi$. Figure 3 (a) shows an overview of our neuron encoder.

**Embedding Layer ($\phi$).** The embedding layer maps input features vector $\mathbf{x}$ into $d$-dim embedding vectors $\tilde{\mathbf{x}}$. It is implemented by a three-layer perceptron with `BatchNorm` and `ReLU` attached to the input and hidden layers. We refer to the outputs of this embedding layer as node embeddings. Formally, we compute $\tilde{\mathbf{x}}_k = \phi(\mathbf{x}_k)$, where $\mathbf{x}_k$ is the $k$-th node of a neuron, or jointly as $\tilde{\mathbf{X}} = \phi(\mathbf{X})$.

**LSTM-based tree-path processor ($\eta$).** The original LSTM [18] architecture applies for strictly sequential data. However, a neuron tree structure is more complicated than a simple sequence. Therefore, we adopt a child-sum Tree-LSTM architecture, similar to [38], to process such data. Recall the hidden state and the cell state at time $t$ and layer $i$ are usually denoted as $\mathbf{h}_{t,i}$ and $\mathbf{c}_{t,i}$ in LSTMs [18]. Here we denote them similarly, but indexed by tree node, as $\mathbf{h}_{k,i}$ and $\mathbf{c}_{k,i}$ at node $k$. To aggregate information from multiple child nodes, we simply sum over the hidden states and cell states. The transition equations can be formulated as followings:

$$\mathbf{h}'_{k,1} = \sum_{j \in C(k)} \mathbf{h}_{j,1}, \quad \mathbf{c}'_{k,1} = \sum_{j \in C(k)} \mathbf{c}_{j,1} \tag{1}$$

$$\mathbf{h}_{k,1}, \mathbf{c}_{k,1} = \text{LSTM}_1(\tilde{\mathbf{x}}_k, (\mathbf{h}'_{k,1}, \mathbf{c}'_{k,1}))) \tag{2}$$

$$\mathbf{h}'_{k,2} = \sum_{j \in C(k)} \mathbf{h}_{j,2}, \quad \mathbf{z}'_k = \sum_{j \in C(k)} \mathbf{z}_j \tag{3}$$

$$\mathbf{h}_{k,2}, \mathbf{z}_k = \text{LSTM}_2(\mathbf{c}_{k,1}, (\mathbf{h}'_{k,2}, \mathbf{z}'_k)) \tag{4}$$

where $C(k)$ denotes the set of child nodes of node $k$, and $\mathbf{z}_k$ is the final cell state. We use two layers of LSTM cells (non-shared weights) to better capture the topology information (Eqns. 2, 4). Our tree-path processor is designed in a bottom-up manner where the information is aggregated from each individual leaf nodes and flowed to the root node. We then use the final cell state of the root node $\mathbf{z}$, referred to as *neuron representation*, for downstream tasks, *e.g.*, predicting class labels by an additional single linear layer classifier or projected by a projection head [5, 16] for contrastive learning. Formally, we compute the representation of a neuron tree by $\mathbf{z} = \eta(\tilde{\mathbf{X}}, \mathcal{G})$ (the propagation path is determined by $\mathcal{G}$). Figure 3 (b) demonstrates how information is propagated along the path.

### 3.3 Contrastive learning of tree graph

Recent progress in self-supervised contrastive learning, stemmed from visual representation learning [45, 16, 6, 5, 12], has been seen in many other fields, including graph contrastive learning [46, 49] — an overall under-explored direction. Attempts have been made to study self-supervised graph representation learning in terms of data augmentation [49], network architecture, and overall pipelines [46]. Although the neuron tree is a special case of graph structure, we argue that it is too special to share the same setups as traditional graphs in terms of contrastive learning [49]. Below we first introduce our TreeMoCo pipeline, then elaborate on the subtle but important difference between a neuron tree and a general graph, subsequently introducing our data augmentations for tree structure.

**Tree Contrastive Learning.** Figure 3 illustrates the overall pipeline of TreeMoCo. The core idea of contrastive learning is to maximize the similarity between positive sample pairs and minimize it among negative sample pairs by optimizing a contrastive loss function [45, 5, 13]. We follow a successful exemplar, MoCo [16, 6], in visual contrastive learning to build our framework.

Conceptually, MoCo [16] abstracts contrastive learning as a dictionary look-up problem. For each encoded query $\mathbf{q}$, there is a set of encoded keys $\{\mathbf{k}_0, \mathbf{k}_1, \mathbf{k}_2, ...\}$ in a dictionary. The task for a model is to pull closer $\mathbf{q}$ and its matched positive key $\mathbf{k}^+$ in the dictionary while to spread $\mathbf{q}$ away from all other negative keys $\mathbf{k}^-$. To that end, we follow the successful experience in [5, 16] to use the dot-product as similarity measurement, and the InfoNCE [45] contrastive loss:

$$L_q = -\log \frac{\exp(\mathbf{q} \cdot \mathbf{k}^+ / \tau)}{\exp(\mathbf{q} \cdot \mathbf{k}^+ / \tau) + \sum_{\mathbf{k}^-} \exp(\mathbf{q} \cdot \mathbf{k}^- / \tau)} \tag{5}$$

where $\tau$ is a temperature hyper-parameter [45]. This loss is defined for one input instance, and the total loss is averaged over a batch.

To construct a pair of query and key, given a set of trees $\mathcal{D}$ and a set of possible augmentations $\mathcal{T}$, we first uniformly sample a tree $\Upsilon \sim \mathcal{D}$ and transform it into two augmented views $\Upsilon_q = t(\Upsilon)$ and $\Upsilon_k = t'(\Upsilon)$, referred to as the query view and the key view, by applying two sampled augmentations $t \sim \mathcal{T}, t' \sim \mathcal{T}$. Then, we compute their neuron representations using a neuron encoder defined in Sec. 3.2, *i.e.*, $\mathbf{z}_q = f(\Upsilon_q), \mathbf{z}_k = f'(\Upsilon_k)$. Following the improvement in SimCLR [6], we further map the neuron representations to another latent space using a non-linear projection head $h(\cdot)$ and normalize them by $\mathbf{q} = h(\mathbf{z}_q)/||h(\mathbf{z}_q)||_2$ and $\mathbf{k} = h'(\mathbf{z}_k)/||h'(\mathbf{z}_k)||_2$. Here, $f'$ and $h'$ stand for the momentum copies of $f$ and $h$, named momentum encoder [16]. Therefore, $\mathbf{q}$ and $\mathbf{k}^+$ in Equation 5 are naturally generated from a single sample, and $\mathbf{k}^-$'s are from a queue [16] that stores features from previous batches.

**Discussion: data augmentation on graphs, trees, and neuron morphology.** Data augmentation is essential for self-supervised contrastive learning [5]. Current successful data augmentations for graphs [49] mainly include (1) "node dropping" that randomly drops graph nodes and their associated edges, (2) "edge perturbation" that randomly adds or drops edges, (3) "attribute masking" that acts like dropout [36] to node features, and (4) "subgraph sampling" that uses random walk to sample a partial graph. These augmentations can apply to general graph data without loss of generality. However, some of them are unsuitable for tree-structure graphs without special consideration. For example, the node dropping at a rate of 0.2 (default in [49]) would produce a pruned tree since all the subtrees would be removed once their parent node was dropped. Besides, edge perturbation is unlikely to be feasible in a tree structure if no strategical design is considered. Two nodes in a neuron tree may be spatially close to each other while having long "distance" in terms of topology (*e.g.*, number of hops from one node to another) and vice versa. The perturbed edges would create cyclic graphs on the one hand, and break the consistency between space (3d coordinates) and structure (nodes connectivity) on the other hand. Therefore, we argue the special consideration is needed for the design of data augmentations on tree-structured data.

**Data augmentation for neuron morphology representation learning.** Regarding the characteristics of neuron data, we propose 11 available augmentations for neuron morphology representation learning, which are briefly summarized in Table 1 and some implementations are provided in Appendix B.3. The rationale behind our design is that neuron representation should be invariant to augmentations — the golden rule applicable in many cases. For neuron data, such augmentations may occur in (i) node coordinates, (ii) overall morphology, (iii) attribute masking and (iv) topology. Thus we can augment neuron data from different perspectives as Table 1 shows. Jittering (4, 6) means adding random noise in a small range; deforming (7) means scaling each branch independently. We also include attribute masking (8) used in general graph augmentation. For (iv) topology, we propose three different augmentations. The first is (9) "truncated dropping subtrees" that recursively visits nodes from the root to leaves and drops a subtree with a small probability (*e.g.*, 0.05). The name "truncated" means at most $m$ revisesubtrees will be dropped. Similarly, we have (10) "skipping parent node" that at most $m$ parent nodes will be replaced by one of their subtrees (respectively), and (11) "swapping sibling child" that picks a subtree from a sibling node and swaps that subtree with another subtree in the current node; all pickings are random. For simplicity, we empirically use the same $m = 10$ for (9)-(11) in this paper. Figure 2 (d) illustrates how topology augmentation acts like.

Table 1: Overview of data augmentations for neuron morphology representation learning.

| Category | Target | Augmentations |
|---|---|---|
| (i) Point Transformation | Coordinates | 1) Scaling, 2) Rotating, 3) Shifting, 4) Jittering, 5) Flipping |
| (ii) Morphology | Branch feats | 6) Jittering branches, 7) Deforming branches |
| (iii) Attribute Masking | Branch feats | 8) Masking attributes |
| (iv) Topology | Edges | 9) Truncated dropping subtrees
10) Truncated skipping parent node
11) Truncated swapping sibling child |

## 4 Experiments

### 4.1 Experiment setups

**Data.** We download data from 3 public resources: (1) BICCN fMOST data from Brain Image Library (BIL, https://download.brainimagelibrary.org/biccn/zeng/luo/fMOST/) [28], (2) Janelia MouseLight (JML, http://mouselight.janelia.org/) [44], and (3) Allen Cell Types (ACT, https://celltypes.brain-map.org/) [1]. In total, 3358 three-dimensional mouse brain neuron reconstructions are collected (refer to Table 2 for details). To facilitate analysis, we exclude cell types with less than 50 reconstructions and the neurons with obvious reconstruction errors, yielding 2403 neurons in total. Some of the samples include both dendrite and axon reconstructions. For simplicity, we only keep dendritic arbors in this paper and leave the analysis of axons for future works. All three datasets have registered reconstructions to the 3D Allen mouse brain Common Coordinate Framework (CCFv3 [41]) and thus their scales are the same. Meanwhile, the absolute coordinates and orientations of registered neurons could also carry cell-type information that are

related to brain anatomy instead of neuron morphology. To eliminate such potential information leakage, all dendrites are shifted and rotated such that the root is placed at the origin and the X/Y/Z coordinates are determined based on principle component analysis (PCA). Data are then processed and analyzed following the steps introduced in the Section 3.1. In the interest of evaluating TreeMoCo's capability of classifying known cell types across data sources, we specifically use the cell types with rich coverage among 3 datasets only (grayed cell types in Table 2). We denote these sub-datasets as BIL-6, JML-4 and ACT, respectively. For supervised training, 80% samples in each dataset are randomly picked as training set and 20% are reserved for testing. For TreeMoCo pre-training, we pre-train on the joint training sets of BIL-6, JML-4, and ACT when evaluating the supervised downstream tasks and train on all samples except those from class "others" when analyzing the embedding space.

Table 2: Demographic table of reconstructions. Refer to Appendix A.2 for full names of brain regions.

| Dataset | Number of neuron reconstructions in each brain regions | | | | | | | | | | | |
|---|---|---|---|---|---|---|---|---|---|---|---|---|
| | Total | L2/3 | L4 | L5 | L6 | VPM | CP | VPL | SUB | PRE | MG | Others |
| BIL [28] | 1741 | 126 | 56 | 315 | 93 | 378 | 311 | 80 | 4 | - | 50 | 328 |
| JML [44] | 1107 | 64 | - | 179 | 114 | 12 | - | 5 | 70 | 60 | - | 602 |
| ACT [1] | 510 | 111 | 123 | 164 | 97 | - | - | - | - | - | - | 15 |
| Total | 3358 | 301 | 179 | 658 | 304 | 390 | 311 | 85 | 74 | 60 | 50 | 945 |

**Implementations and evaluation.** We postpone the detailed implementations and settings of hyper-parameters of TreeMoCo to Appendix B.1. For implementations of all the compared methods, we directly use the official codes and the default settings of MorphVAE [22], TRNN [52] and GraphCL [52] to obtain the results. Please refer to Appendix B.2 for their implementation details. All models share the same 29-d input features and are trained for 100 epochs. For evaluation, we report the sample-wise accuracy. During pre-training, we evaluate models every 5 epochs via a K Nearest-Neighbor (KNN) classifier for unsupervised methods or their inherent classification heads for supervised methods and report the best test set performance for all the studied methods. This strategy mainly follows [45, 49]. We set the number of neighbors to 20 for BIL-6 and ACT datasets and 5 for the JML-4 dataset since a minority class might only have 12 samples in total in JML-4.

## 4.2 TreeMoCo delineates neuron morphology in the embedding space

**Visual inspection of cell types delineation.** We first demonstrate that TreeMoCo can learn neuron tree representations that manifest different characteristics of neuron morphology. Self-supervised training is conducted on the full dataset, and the distribution of the embeddings is visualized by t-SNE [39] in Figures 1 and 4. By visual inspection, it is evidenced that TreeMoCo can separate major cell-types such as pyramidal neuron (Figs. 1(b-c)) and interneuron (Figs. 1(d-e)). Pyramidal neurons are the primary long-range output neurons of the brain [35] while interneurons mainly sculpt local network dynamics through a variety of circuit motifs. The asymmetric long projecting apical tuft is a unique feature that differentiates pyramidal neuron's morphology with interneuron's. TreeMoCo can further delineate subtypes of pyramidal neuron and interneurons. In Figure 1 and 4(a), we can clearly see the separation of layer 2/3 (Fig. 1(c)) and layer 5 (Fig. 1(b)) pyramidal neurons. Layer 2/3 neurons can be distinguished by their shorter apical dendrites with extensive oblique branching [43] while layer 5 neurons feature extended apical dendrites with extensive horizontal branching in layer 1 and sparse oblique branching in layer 2/3 [14]. The two primary classes of inhibitory interneurons (parvalbumin-expressing basket cells and somatostatin-expressing Martinotti cells) are also clearly distinguished with TreeMoCo. Parvalbumin-expressing basket cells have extensive branching field that extends radially around the soma (Fig. 1(d)) [29]. In contrast, somatostatin-expressing Martinotti cells tend to have unipolar or bipolar dendritic branching that extends vertically above and below the soma (Fig. 1(e)) [34]. It should be noted that such delineation is purely self-driven without any human labels or any prior knowledge of cell types. In contrast, the delineation is less clear in GraphCL's result and not observed in MorphVAE's result (details in Appx. D).

**Quantitative evaluation.** Here we quantitatively evaluate the quality of learned representations of different methods in classifying the known cell-types. For supervised methods, we directly train them on each dataset. For unsupervised methods, we pre-train them on the joint training sets. Table 3 reports the results, with the best test KNN accuracy during unsupervised pre-training or the best test

Table 3: Comparisons to state-of-the-arts neuron cell-type classification. We report the sample-wise accuracy (%). The numbers of classes in BIL-6, JML-4 and ACT are 6, 4, and 4, respectively. [†] Results obtained from officially released codes.

| Methods | BIL-6 | JML-4 | ACT |
|---|---|---|---|
| *Supervised training in individual datasets.* | | | |
| [†]MorphVAE [22] | 66.80 | 40.00 | 41.05 |
| [†]TRNN [52] | 76.56 | 51.43 | 46.32 |
| *Unsupervised pre-training on joint training sets.* | | | |
| [†]GraphCL [49] frozen KNN | 69.14 | 54.29 | 58.95 |
| TreeMoCo frozen KNN | 78.91 | 64.29 | 55.79 |
| TreeMoCo fine-tuned | 83.98 | 67.14 | 68.42 |

Table 4: Ablation on augmentations. We report the accuracy (%) of KNN on BIL-6 test set with models pre-trained on the full BIL-6 dataset with different augmentations.

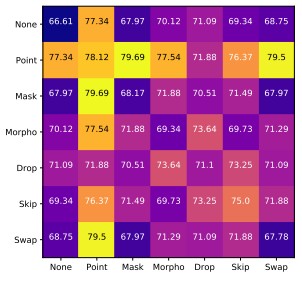

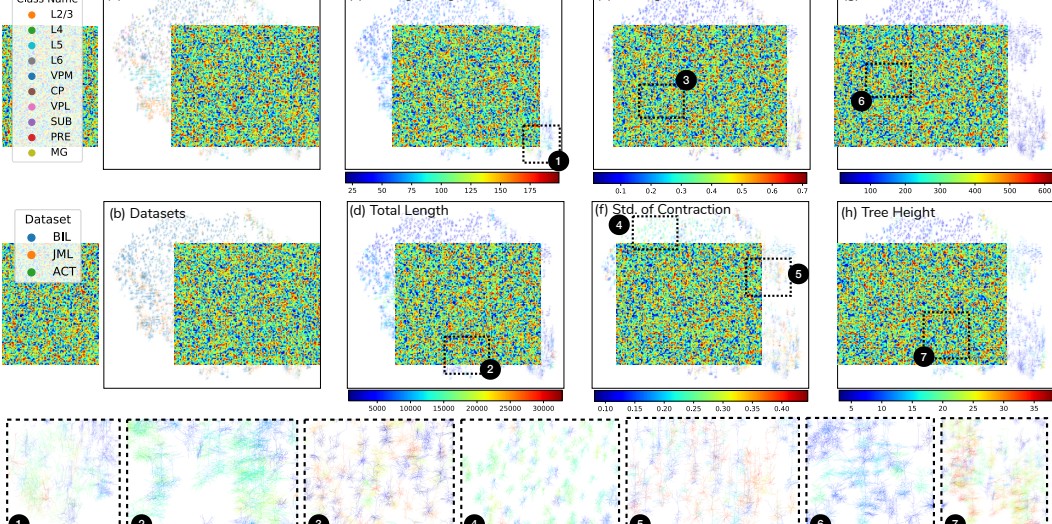

Figure 4: Thumbnails of neurons in the same embedding space of Figure 1 colored by different attributes. It is evidenced TreeMoCo encodes cell types (a) and neuron morphology features (c)-(h), *without human labels*. Refer to Appendix D for high-resolution figures.

accuracy during supervised training included. It can be observed that TreeMoCo trained without label information can already achieve better performance than supervised MorphVAE [22] and TRNN [52]. Fine-tuning the pre-trained neuron encoder further improves cell-type classification. This is mainly because MorphVAE [22] is based on random walk to capture topology information, which ignores tree architecture. While TRNN [52] has less description power than our neuron encoder in capturing high-order interaction between features. GraphCL [49] outperforms TreeMoCo in the ACT dataset but falls behind in BIL-6 and JML-4 datasets. This coincides with the data quality differences (ACT reconstruction is less complete than BIL and JML, more details in Appx. A.2). One explanation is that GraphCL pays more attention to local context and thus is less sensitive to global changes. It should be emphasized that although the overall test set performance is improved during training, we observe that the test KNN accuracy of both TreeMoCo and GraphCL fluctuates and suffers variations, especially for the smaller JML-4 and ACT datasets. GraphCL fails to improve models over time in the JML-4 dataset under some scenarios. Please refer to Appendix C for more details and discussions, where we provide the performance curves of different methods over training.

**TreeMoCo manifests neuron morphology distribution.** Figure 4 shows the t-SNE plots of neuron representations. Each plot is colored by a different morphology feature, *i.e.*, average/total branch length (c, d), average / standard deviation of contraction (e, f), number of nodes in a neuron (g) and the height of a neuron tree (h). Neurons that share similar morphology features are well clustered in

the embedding space (see (1) to (7)), demonstrating the good representations learnt by TreeMoCo. These morphology attributes have not been explicitly provided as input features but it turns out that TreeMoCo can mine such information by itself, *without human labels*. In addition, we can see it by cross-referencing Figure 4 (a) and (b) that neurons of a same class but from different datasets are clustered together in TreeMoCo's representation space, showing TreeMoCo's unbiasedness to data sources — a much desirable property for neuron morphology representation learning.

### 4.3 Ablation on augmentation methods and network architectures

In this subsection, we first assess the importance of data augmentations in contrastive neuron morphology learning. Then we ablate on several design choices in our neuron encoder.

To systematically study the effects of different categories of data augmentations, we first group the proposed augmentations in Table 1 by their categories, except for the topology augmentation, the one we are more interested in. Then, we apply them individually or in pairs to TreeMoCo framework with a higher applying probability due to the reduced number of augmentations (details in Appx. B.2.) when pre-training. Note that this ablation does not aim to figure out the best compositions of presented augmentations, but to unravel how different augmentations contribute to the final performance. For ablation purposes, in this experiment we train on the full BIL-6 dataset for 50 epochs. Table 4 shows the KNN evaluation results under each individual category or a composition of data augmentations. Here we do not specify the order of augmentations, thus resulting a symmetric performance matrix. Unlike the observation in visual or graph representation contrastive learning that no single augmentation performs the best [5, 49], we find in neuron morphology representation learning certain kinds of augmentations are good enough for pre-training (*e.g.*, point transformation, skipping parent node). Topology augmentations do not necessarily bring better performance. This could be caused by the mechanism of Tree-LSTM, which aggregates the whole tree to generate embeddings and thus is sensitive to the global changes caused by topology augmentation. Better neuron morphology representations might be excepted if a more proposer set of augmentations or a better network architecture were applied under the TreeMoCo framework.

Then, we study the effects of different components in our neuron encoder. For this experiment, we train neuron encoders for 100 epochs on three datasets and report their best test performance. Table 5 (left) shows the results, where a steady increasing trend of model performance can be observed. We also include the KNN accuracy (Tab. 5 right) for 50-epoch pre-training on the joint dataset and evaluated in BIL-6 dataset just for ablation interest. Here we find that batch normalization (BN) is beneficial for both contrastive self-supervised learning and supervised learning. No BN in the neuron encoder would give significantly worse results (first line of right table). Further adding BN in the projection head is helpful for TreeMoCo. This behavior has also been reported in non-contrastive SSL methods before, *e.g.*, BYOL [12, 31], where removing BN causes model collapse.

Table 5: **Ablation on model designs. (left)** Classification accuracy of supervised training for 100-epoch. **(right)** KNN accuracy of TreeMoCo pre-training for 50 epochs and evaluated in BIL-6 dataset. BN matters more in TreeMoCo. Gray rows indicate our default setting.

| Settings | | | Classification Accuracy (%) | | |
|---|---|---|---|---|---|
| BN | MLP | # LSTM Cells | BIL-6 | JML-4 | ACT |
| - | - | 1 | 80.86 | 54.29 | 43.16 |
| - | ✓ | 1 | 83.98 | 57.14 | 50.53 |
| ✓ | ✓ | 1 | 84.77 | 58.57 | **63.16** |
| ✓ | ✓ | 2 | **85.55** | **64.29** | 62.11 |

| Encoder BN | Projector BN | KNN accuracy (%) BIL-6 |
|---|---|---|
| - | - | 69.14 |
| ✓ | - | 75.00 |
| ✓ | ✓ | **76.56** |

## 5 Conclusion and Limitations

In this work, we have explored contrastive learning for neuron morphology representation learning via TreeMoCo. Our results on 2403 neuron reconstructions show that TreeMoCo has the unique potential to identify unbiased cell-types. Modern cell-type classification is heavily dependent on transcriptomic data [50, 8]. However, even neurons within the same mouse possessing identical genetic identity and birth-date demonstrate extremely variable dendritic morphology and synaptic distribution [19]. This variability underlies the need for high-throughput neuronal morphology classification techniques to probe for the existence of novel subtypes within classically established

neuronal identities [3]. TreeMoCo has a great potential to fill this need and to explore dendritic architecture. These findings could lead to more rigorous output measurements for drug discovery experiments and aid in the development of treatments for neurological disorders characterized by progressive dendrite degeneration such as Alzheimer's disease and depression [10, 7].

We note that TreeMoCo is still a preliminary effort of its kind. Certain limitations can be observed and need to be solved in the future. For instance, Tree-LSTM aggregates the whole graph and thus is sensitive to global changes. This results in its performance decay on incomplete neuron reconstruction (ACT dataset) and its favor of node-feature-based augmentation over topology-based augmentation. Moreover, compared to most image contrastive learning frameworks, TreeMoCo's performance on downstream tasks could oscillate after long training epochs on small datasets. An effective early stopping criterion is needed but absent when label information is not available.

Nevertheless, the tree graph is a common data structure representing objects varying from artery and lung vessels [21] to program source code [26] and HTML documents [4]. Another direction of our future work is to extend TreeMoCo into a more general framework for learning representations beyond neuron morphology. Following the success of contrastive learning in image analysis problems, we believe TreeMoCo could be a powerful tool to solve unsupervised and few-shot learning problems related to a wide range of tree graph data.

## Acknowledgments and Disclosure of Funding

Daniel Maxim Iascone is sponsored by NIH (F32 MH125600-01).

Lijuan Liu is sponsored by a MOST (China) Brain Research Project, "Mammalian Whole Brain Mesoscopic Stereotaxic 3D Atlas" (2022ZD0205200, 2022ZD0205204).

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
