# Appendix to TreeMoCo: Contrastive Neuron Morphology Representation Learning

**Hanbo Chen**[*,†]
Tencent AI Lab
hanbochen@tencent.com

**Jiawei Yang**[*,†]
University of California, Los Angeles
jiawei118@ucla.edu

**Daniel Maxim Iascone**
University of Pennsylvania, Philadelphia
daniel.iascone@pennmedicine.upenn.edu

**Lijuan Liu**
Southeast University, Nanjing
juan-liu@seu.edu.cn

**Lei He**
University of California, Los Angeles
lhe@ee.ucla.edu

**Hanchuan Peng**
Southeast University, Nanjing
h@braintell.org

**Jianhua Yao**[‡]
Tencent AI Lab
jianhuayao@tencent.com

## A    Additional information of neuron data and features

### A.1    Limitations of handcrafted features

In this section, we will show some examples of neuron branches and neuron dendrites that cannot be differentiated by handcrafted features.

**Limitations of branch features.**    We take a sequence of vertices between two bifurcations or a bifurcation (fork node) and a tip node (leaf) as a branch. A branch can be viewed as a sequence of vectors $E = [\vec{E_0}, \vec{E_1}, ..., \vec{E_{n-1}}]$, where $\vec{E_i} = V_i - V_{i+1}$ is the vector pointing from a child node to its parent node. Two commonly used measurements measuring the shape of a branch are *length* and *contraction*. The formulas of these two metrics are detailed in Sec. 3.1 and Figure A1 (a). Here, we elaborate on the limitations of both metrics. Based on the definition of length and contraction, it is evident that both metrics are invariant to rotation and shifting of the branch — two desired properties. What is less obvious is that both metrics are invariant to the order of branch vectors $\vec{E_i}$. As shown in Figure A1 (a), by only re-ordering of edge vectors, a branch's shape can be significantly changed from a bow shape to a zig-zag shape while its length and contraction remain the same.

**Limitations of neuron features.**    Most classic neuron morphology analysis works adopt the mean and the standard deviation of handcrafted features related to branches or bifurcations to measure the shape of a neuron. Such measurement is taken in a bag-of-word (BOW) fashion. The context and spatial relationship between arbors are ignored under this measurement. Figure A1 (b) shows that the neurons of significantly different shapes could share similar features based on this kind of measurement. In this paper, our proposed neuron encoder can capture information from both the branch features as well as the context and spatial relation via the interaction between the embedding layer and the tree-LSTM-based path processor.

---

[*]: Equal contribution. [†]: Work done when authors were at Tencent AI Lab. [‡]: Corresponding author.

36th Conference on Neural Information Processing Systems (NeurIPS 2022).

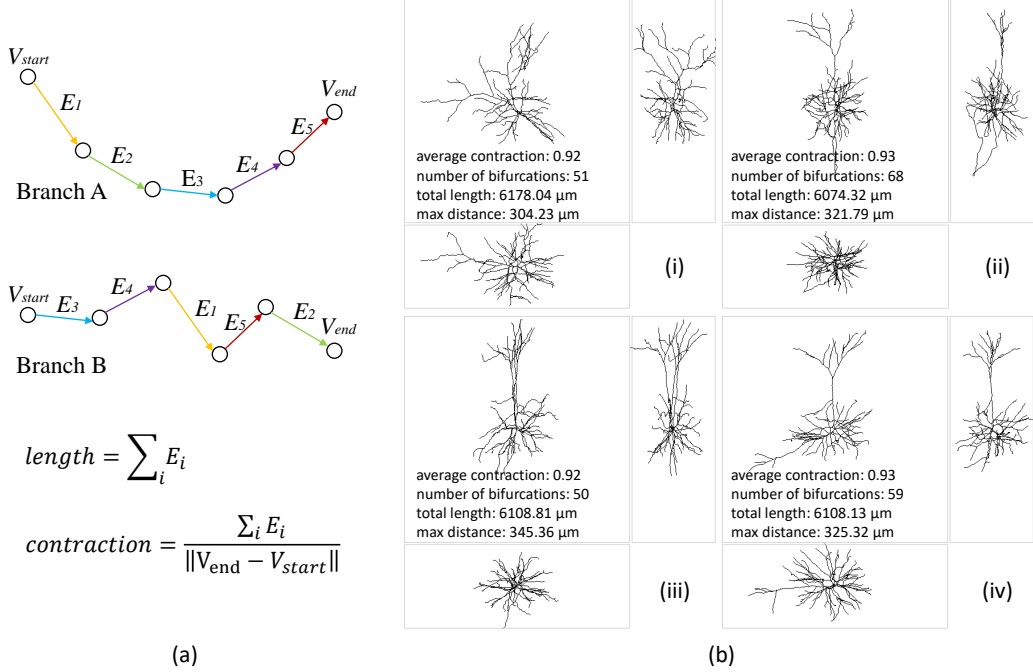

Figure A1: Illustration of limitation of handcrafted features. (a) Length and contraction are invariant to the order of edge vectors. Branch A and B are composed of the same set of edge vectors. They are of different shapes but share the same length and contraction value. (b) Neurons of different shapes may have similar features. Each neuron is viewed from three different angles. Feature values are listed in the subplot.

## A.2 Dataset details

Three public datasets are used in this study. All of them contain 3D neuron reconstructions from mouse brains. Reconstructions are registered to Allen mouse brain Common Coordinate Framework (CCFv3 [11]) in their original release. The cell type of a neuron is assigned based on its brain anatomical region. This information comes together with each dataset. Note that the labels of samples from the JML dataset are purely determined by the registration results without manual inspection. In contrast, in ACT and BIL datasets, such information has been visually inspected and corrected by neuroscientists and thus is more accurate. For JML and BIL datasets, whole neuron reconstructions are generated, including the long-range projection of axons. Due to the limitation of imaging techniques, the ACT dataset's neuron reconstructions are limited to a thin 3D slice of brain tissue and thus not complete. This type of data with incomplete morphology is typical in neuroscience studies due to the technical limitations and cost efficiency concerns. Thus, to show if and to which degree our proposed method could work on incomplete reconstruction, we keep the ACT dataset for analysis even if it is incomplete. To summarize, the BIL dataset has the best quality – complete 3D reconstructions and reliable cell type labels. More details of the datasets can be found in Table A1.

To facilitate analysis, we merge the isocortex cell types by layers and only keep 10 cell types with relatively broad coverage. The abbreviations and full names of these brain regions are listed in Table A2. Examples of neuron reconstructions of different types from different datasets can be found in Figures A2 and A3. It is visually evident that ACT's reconstructions are incomplete compared to BIL's and JML's. Also, with a visual impression, the dendritic morphology of some cell types, such as VPM, VPL and CP, are similar and thus challenging to be differentiated.

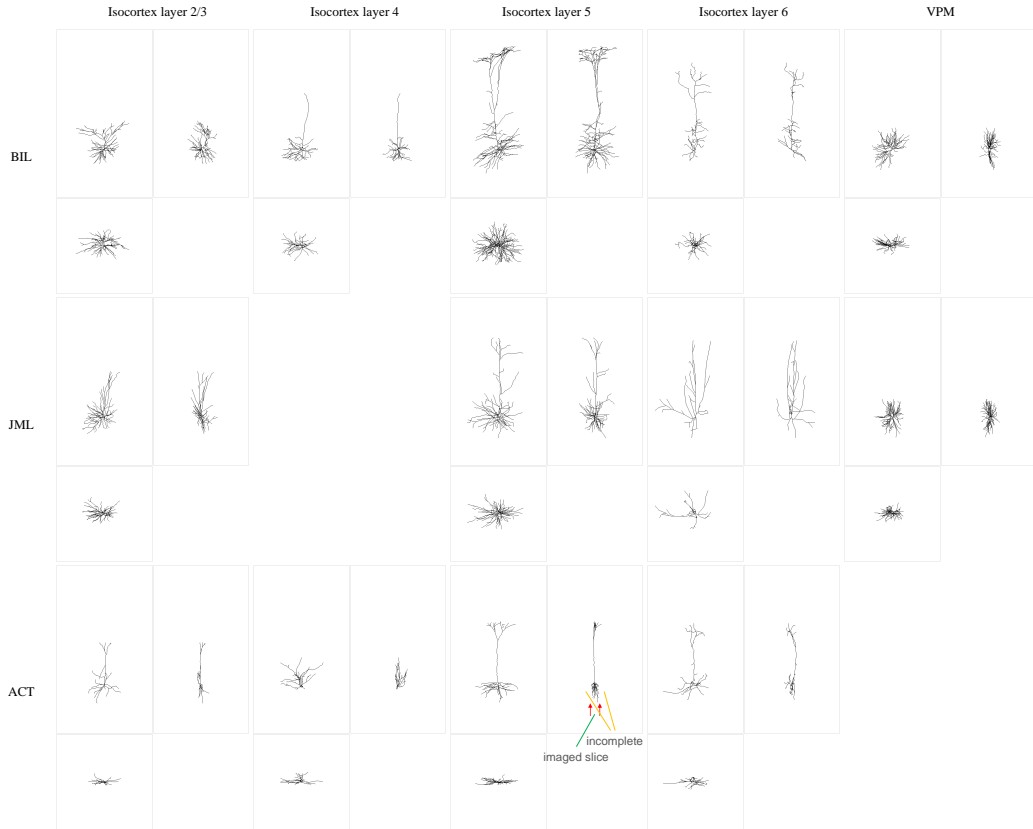

Figure A2: Example reconstructions of 5 neuron cell types that are common between datasets. Each column corresponds to a cell type. Each row corresponds to a dataset. Each subplot shows 3 different views (three-view) of the same neuron – top-left: front view, top-right: side view, bottom-left: top view. Thumbnails are taken on the same scale for all neurons. As evident from the side view, ACT's reconstructions are all flat compared to compared to BIL's and JML's. It is because ACT reconstructions are from a thin tissue slice and thus incomplete.

## B    Implementation details

Here we elaborate on the implementation details. Our code and pre-trained models will be made public once the paper is accepted.

### B.1    Implementation details of TreeMoCo training.

**Model architectures.**    For implementation, we set the layer sizes of embedding MLP to 256 and 128, and the cell state size of LSTM-processor to 128. Our detailed model architectures in PyTorch style are:

- Embedding layer $\phi$:
  ```
  embedding = Sequential(
              Linear(29,128), BatchNorm(128), ReLU(),
              Linear(128,256), BatchNorm(256), ReLU(),
              Linear(256,128))
  ```

- LSTM-based tree path processor $\eta$:
  ```
  W1_iouf, U1_iouf = Linear(128,512), Linear(128,512) # first cell
  W2_iouf, U2_iouf = Linear(128,512), Linear(128,512) # second cell
  ```

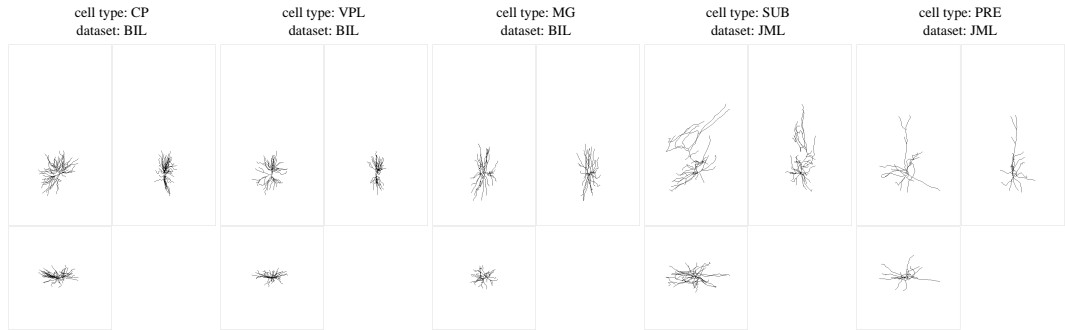

Figure A3: Example reconstructions of 5 neuron cell types that are only present in one dataset. Each subplot shows 3 different views (three-view) of the same neuron – top-left: front view, top-right: side view, bottom-left: top view. Thumbnails are taken on the same scale for all neurons.

Table A1: Details of datasets used in this paper.

| Dataset | BIL | JML | ACT |
|---|---|---|---|
| Full Name | BICCN fMOST data from Brain Image Library | Janelia Mouse Light | Allen Cell Types |
| Reference | [8] | [12] | [1] |
| Total reconstructions | 1741 | 1107 | 510 |
| Reconstructions used | 1413 | 505 | 495 |
| Complete reconstruction | Yes | Yes | No |
| Cell type label | Manual | Auto | Manual |
| Tissue labeling | TIGRE2.0 transgenic reporter lines coupled with Cre espression | A mixture of low-titer AAV Syn-iCre and a high-titer Cre-dependent reporter | Biocytin-filled |
| Image modality | fMOST | MouseLight | flourescent and brightfield |
| Reconstruction method | Automatic reconstruction followed by manual correction and automatic refinement | Automatic reconstruction followed by manual correction | Automated 3D reconstruction manually curated using the Mozak extension of Vaa3D. |
| Registration method | mBrainAligner | LandMark | Manual |

Links for datasets:
  BIL: https://download.brainimagelibrary.org/biccn/zeng/luo/fMOST/
  JML: http://mouselight.janelia.org/
  ACT: https://celltypes.brain-map.org/

- Projection head $h$:
```
projector = Sequential(
            # non-linear projection head
            Linear(128,128), BatchNorm(128), ReLU(),
            Linear(128,128))
```

**LSTM processor forward.** We provide a pseudo-code for LSTM forward in Algorithm 1. Our implementation is based on Deep Graph Library (DGL) [10], an efficient library for graph propagation in deep learning.

**Pre-training details.** We provide a pseudo-code for TreeMoCo training in Algorithm 2. We train TreeMoCo with a batch size of 128 in one V100 GPU. We use an SGD optimizer with a weight decay of 5e-4 and a constant learning rate of 0.06. The temperature parameter $\tau$ is 0.1, the momentum parameter to update the momentum encoder is 0.99, and the queue length to store negative keys is 1024. We use a smaller queue than MoCo [4] (1024 v.s. 65536) since the total number of neurons is significantly less than the number of images in visual representation learning [4, 3] (2.3k v.s.

Table A2: Abbreviation of brain region and cell types. All brain regions are defined following Allen Reference Atlas. Refer to the website for more details: `http://atlas.brain-map.org/atlas?atlas=602630314`

| Abbreviation | Full name |
|---|---|
| l2/3 | isocortex layer 2/3 |
| l4 | isocortex layer 4 |
| l5 | isocortex layer 5 |
| l6 | isocortex layer 6 |
| VPM | Ventral posteromedial nucleus of the thalamus |
| VPL | Ventral posterolateral nucleus of the thalamus |
| CP | Caudoputamen |
| SUB | Subiculum |
| PRE | Presubiculum |
| MG | Medial geniculate complex |

---

**Algorithm 1** LSTM-based tree path processor ($\eta(\cdot)$) forward: PyTorch-like Pseudocode

---

```
# W_iouf, U_iouf: weights for input features and hidden states for a LSTM cell.
# nodes: data structure to store graph and data, used in DGL.
# embedding layer
feats = embedding(feats)
# initial states
nodes.data['iouf'] = W1_iouf(feats)
nodes.data['h1'], nodes.data['c1'] = zeros((N,128)), zeros((N,128))
nodes.data['h2'], nodes.data['z'] = zeros((N,128)), zeros((N,128))

def propagate(nodes):
    # nodes: [N, j, C], currently processing N nodes, each of which has j subtrees that own C-dim
        representations.
    h1, c1 = nodes.mailbox['h1'], nodes.mailbox['c1']
    h2, z = nodes.mailbox['h2'], nodes.mailbox['z']
    # sum over child nodes
    h1, c1, h2, z = h1.sum(-2), c1.sum(-2), h2.sum(-2), z.sum(-2)
    # first LSTM cell
    xi, xo, xu, xf = chunk(nodes.data['iouf'], 4, 1)
    hi1, ho1, hu1, hf1 = chunk(U1_iouf(h1), 4, 1)
    i, f, g, o = sigmoid(xi + hi1), sigmoid(xf + hf1), tanh(xu + hu1), sigmoid(xo + ho1)
    c1 = i * g + f * c1
    h1 = o * tanh(c1)

    # second LSTM cell
    xi, xo, xu, xf = chunk(W2_iouf(c1), 4, 1)
    hi2, ho2, hu2, hf2 = chunk(U2_iouf(h2), 4, 1)
    i, f, g, o = sigmoid(xi + hi2), sigmoid(xf + hf2), tanh(xu + hu2), sigmoid(xo + ho2)
    z = i * g + f * z
    h2 = o * tanh(z)
    return {'h1': h1, 'c1': c1,
            'h2': h2, 'z': z}
```

---

1.28M). For data augmentations, we *have not searched* their combinations and hyper-parameters extensively. Therefore, our default setting of data augmentations may not be optimal for pre-training. We emphasize that even under our current naive combination of data augmentations, TreeMoCo has shown promising performance, showing its potential for gaining improved representations with better combined and parameterized data augmentations used during pre-training. The default data augmentations we use for pre-training are:

```
RandomScaleCoordsBranches(p=0.2, scales=[0.8, 1.2]),
RandomRotate(p=0.5),
RandomJitter(p=0.2, sigma=1, clip=5),
RandomShift(p=0.2, shift=[0.2, 0.2, 0.2]),
RandomFlip(p=1),
RandomMaskFeats(p=0.2),
RandomJitterBranches(p=0.2, sigma=0.1, clip=1),
RandomDeformation(p=0.2, scales=[0.8, 1.2]),
RandomDropSubTrees(p=0.05, max_cnt=10),
```

```
                RandomSkipParentNode(p=0.05, max_cnt=10),
                RandomSwapSiblingSubTrees(p=0.05, max_cnt=10)
```

**Evaluation details.**   During pre-training, we extract features from the backbone and the projection head and evaluate them individually with the KNN protocol. We report the best KNN accuracy between them. Note that, in visual representation learning, the features extracted from the backbone are usually better than those from the projection head in terms of clustering performance [2, 14]. However, here we find the best KNN accuracy comes from the backbone for the BIL-6 dataset and the projection head for JML and ACT datasets. We leave the discussion and analysis for such an observation to future work. For fine-tuning evaluation, we initialize the neuron encoder with pre-trained weights and train it for 100 epochs using the Adam optimizer [5] with a learning rate of 2e-3 and a weight decay of 0.02. The projection head is removed and unused when fine-tuning.

---

**Algorithm 2** TreeMoCo training: PyTorch-like Pseudocode

---

```
# f_q, h_q: encoder: backbone, projection head
# f_k, h_k: momentum encoder: momentum copies of backbone and projection head
# m: momentum coefficient
# tau: temperature

for batch_neuron in loader: # load a minibatch x with N samples
    x_q, x_k = aug(batch_neuron), aug(batch_neuron) # augmentation
    z_q, z_k = f_q(x_q), f_k(x_k) # positive pairs (query, key): [N, C] each
    q, k = h_q(z_q), h_k(k) # projection: [N, 128] each
    q, k = normalize(q), normalize(k) # l2 normalization

    logits_pos = dot(q, k) # similarity between positive pairs: [N, 1]
    logits_neg = mm(q, queue.t()) # similarity bewteen negative pairs: [N, K]
    logits = cat([logits_pos, logits_neg], dim=1) # Nx(1+K)
    labels = range(N)
    loss = CrossEntropyLoss(logits/tau, labels)
    queue.dequeue_and_enqueue(k) # update queue
    loss.backward()
    update([f, h]) # optimizer update: f, h
    [f_k, h_k] = m*[f_k, h_k] + (1-m)*[f_q, h_q] # momentum update: f_k, h_k
```

---

**Notes**: `dot` is vector dot-product. `mm` is matrix multiplication. `k.t()` is k's transpose. `f_q`, `f_k`, `h_q` and `h_k` corresponds to $f, f', h$ and $h'$ in the main text respectively.

## B.2   Implementation details of compared methods

**MorphVAE.**   For MorphVAE [6], we adopt the officially released code [1] and use its default settings to obtain results. Instead of only using the x/y/z coordinate of nodes, we use the same set of processed neurons and 29-d features. MorphVAE is based on the variational auto-encoder (VAE) that aims to reconstruct from noisy inputs and further combines a classification head to utilize label information. Since MorphVAE is a path-based method that requires the "random walk" to sample paths from leaf nodes to the root node, we follow the original work to sample 256 paths with a maximum length of 32 from leaf nodes to the root node for each processed neuron tree reconstruction. The best testing set accuracy during 100-epoch training (evaluated every 5 epochs as other methods) is reported for a fair comparison. TRNN [15] and our neuron encoder encode the whole tree structure, while MorphVAE encodes the sampled paths inside the trees, which could ignore the tree topology information. And thus, it is not surprising that the performance of TRNN and our TreeMoCo is superior to MorphVAE (Table 3).

**TRNN.**   For TRNN [15], we follow the officially released code [2] and use its default settings to obtain results. We change the input layer of the TRNN to take our processed tree graph with 29-d node features as inputs. Similar to evaluations of other methods, we monitor the test set performance every 5 epochs for 100 epochs during training and report the best test accuracy.

**GraphCL**   For GraphCL [13], we use the officially released code[3] to obtain the results. We first convert our dataset to the TU dataset format [7] and directly use their default settings for unsupervised

---

[1]`https://github.com/berenslab/morphvae`
[2]`https://github.com/thomasaimondy/treestoolbox/tree/master/casia`
[3]`https://github.com/Shen-Lab/GraphCL/tree/master/unsupervised_TU`

TU training. The input features are the same as other methods, *i.e.*, 29-d features. We train GraphCL for 100 epochs on the joint training sets, evaluate the KNN test accuracy every 5 epochs, and report the best one. In fact, the original setting of GraphCL code is to evaluate models every 10 epochs by training a linear SVM classifier on the training set and testing on the test set. Then, the best performance on the test set is reported. Here, we change it to the KNN evaluation protocol and the evaluation interval from every 10 epochs to every 5 epochs. The official code suggests choosing from three sets of augmentations (refer to the link for more details). We provide the results obtained from them in Table B1. We report the results under the "Random2" augmentation set in the main paper (Tab. 3) since it achieves the best results on two datasets.

Table B1: Frozen KNN evaluation results of GraphCL under different compositions of data augmentations. We report the sample-wise accuracy (%).

| Augmentation Set | Detailed Augmentations | BIL-6 | JML-4 | ACT |
|---|---|---|---|---|
| *Unsupervised pre-training on joint training sets.* | | | | |
| Random2 | NodeDrop, Subgraph | **69.14** | 54.29 | **58.95** |
| Random3 | NodeDrop, Subgraph, EdgePert | 67.58 | 57.14 | 57.89 |
| Random4 | NodeDrop, Subgraph, EdgePert, AttrMask | 67.58 | **61.43** | 55.79 |

## B.3 Implementation details of data augmentation

**Data augmentations in pre-training.** A detailed overview of data augmentations is provided in Table B2. Most of the implementations can be inferred by names. Particularly, for (1) `RandomScaleCoordsBranches(p=0.2, scales=[0.8, 1.2])`, we sample a scale factor from `[scales[0], scales[1]]` uniformly, and multiply $[x, y, z; L, F_E]$ (refer to the last line in Sec. 3.1) with the scale factor; for (2) rotation, we uniformly sample three angles from $[0, 2\pi]$ for $x$, $y$, and $z$, respectively and do a rotation with those angles; for (3) and (4), shifting means the translations are isotropic — three shifting scalars are sampled for $x$, $y$, and $z$ and added to all the point coordinates, while jittering means the translations are anisotropic — each point has its own shifting scalars. The (4) jittering process is: `jitters = np.clip(sigma * np.random.randn(coords.shape[0], 3), clip, clip); coords += jitters`; this process applies similarly to (6) but its targets are branches $F_E$. For flipping (6), the input will be flipped alongside the $x$ and the $y$ axes with 4 possibilities. For (7) deformation, a scale vector of the same shape as $F_E$ will be uniformly sampled between the given range and be applied to $F_E$. For (8) random masking, a percentage of $(p * 100)\%$ features from $F_E$ will be set to zero. For (9) to (11), please cross-reference to the descriptions in Section 3.3 (Data augmentation for neuron morphology representation learning), the illustrations in Figure 2 (d) and the python-style pseudo-codes in Algorithm 3.

Table B2: Overview of data augmentations for neuron morphology representation learning.

| Category | Target | Augmentations |
|---|---|---|
| (i) Point Transformation | Coordinates | 1) `RandomScaleCoordsBranches(p=0.2, scales=[0.8, 1.2])`, 
 2) `RandomRotate(p=0.2)`, 
 3) `RandomShift(p=0.2, shift=[0.2, 0.2, 0.2])`, 
 4) `RandomJitter(p=0.2, sigma=1, clip=5)`, 
 5) `RandomFlip(p=1)` |
| (ii) Morphology | Branch feats | 6) `RandomJitterBranches(p=0.2, sigma=0.1, clip=1)`, 
 7) `RandomDeformation(p=0.2, scales=[0.8, 1.2])` |
| (iii) Attribute Masking | Branch feats | 8) `RandomMaskFeats(p=0.2)` |
| (iv) Topology | Edges | 9) `RandomDropSubTrees(p=0.05, max_cnt=10)`, 
 10) `RandomSkipParentNode(p=0.05, max_cnt=10)`, 
 11) `RandomSwapSiblingSubTrees(p=0.05, max_cnt=10)` |

**Data augmentations in ablation study.** In the ablation study on data augmentations (Sec. 4.3), we use the data augmentations in different categories individually or in pairs. Since the number of augmentations is reduced, we increase the applying probabilities of (6, 7, 8) in Table B2 from 0.2 to 0.5 to better inspect their influence. Point transformations' applying probabilities are not changed since they are a combination of five augmentations.

# C   Learning curves

Under our evaluation protocol, all methods' best test set performance is reported. However, the performance evaluated in small datasets like ours usually fluctuates. To get a complete view and the stability of different methods, we provide their performance curves here. Figure C1 shows the KNN test set accuracy during training (recorded every 5 epochs). The dashed lines of the same color denote the linear trend lines of different curves. Please refer to Appendix B.2-GraphCL and Table B1 for the meaning of "random2/3/4" in Figure C1. The sub-figures' y axes are aligned so that the curves can be directly compared.

Despite suffering variance, the overall performance in the BIL-6 dataset is improved over training in all four sub-figures (TreeMoCo and GraphCL). The performance fluctuation of JML-4 and ACT datasets is larger than that of BIL-6. The dataset size matters here since the number of test set samples would affect the granularity of accuracy intervals, exhibiting different degrees of variance. JML-4 and ACT datasets are much smaller than the BIL-6 dataset (about 1.2k/0.3k/0.3k samples for BIL-6, JML-4, and ACT datasets, respectively) and therefore suffer more performance variance. Collecting more data could alleviate this issue. Regarding stability, TreeMoCo and GraphCL perform less stable in the JML-4 and ACT datasets. However, TreeMoCo can still improve the accuracy in the JML-4 dataset over training. In contrast, GraphCL fails to improve that with longer training.

With observations from Figure C1, we emphasize the particularity of the neuron tree morphology learning problem and the datasets. Existing large-scale benchmarks in visual or lingual domains usually have abundant training/testing samples and thus can evaluate models precisely with minor variance. However, neuron morphology learning is a new ground to explore. Current insufficient testing samples might make the evaluation noisy and less accurate. We hope this early work could contribute to future exploring in designing a more reliable evaluation protocol.

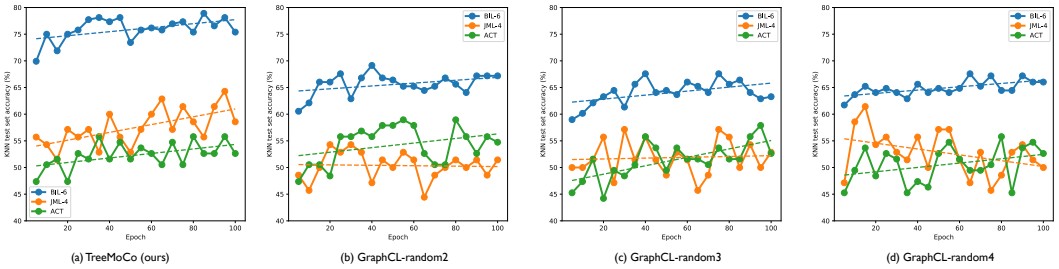

Figure C1: **KNN test set accuracy during training.** The dashed lines denote the linear trend lines.

# D   t-SNE plot of different self-supervised methods

We train models on the full dataset without "others" for 100 epochs to analyze and compare the embeddings learned by different self-supervised methods. The distribution of the same set of neurons in the embedding space is then visualized by projecting the encoded representations into a 2D space with t-SNE [9]. The visualization results are shown in Figures 1 and 4 and below (Figs D1 and D2).

## D.1   t-SNE plot of MorphVAE

When training MorphVAE in a self-supervised fashion, the classification loss is set to zero, and only the reconstruction loss is computed. The resulted t-SNE plot is shown in Figure D1. It is obvious that MorphVAE largely fails to generate meaningful embedding to differentiate neuron morphology. This is in consensus with the observation in the original MorphVAE paper [6] that without a classification loss to train the encoder, MorphVAE's performance on downstream classification tasks will significantly degenerate.

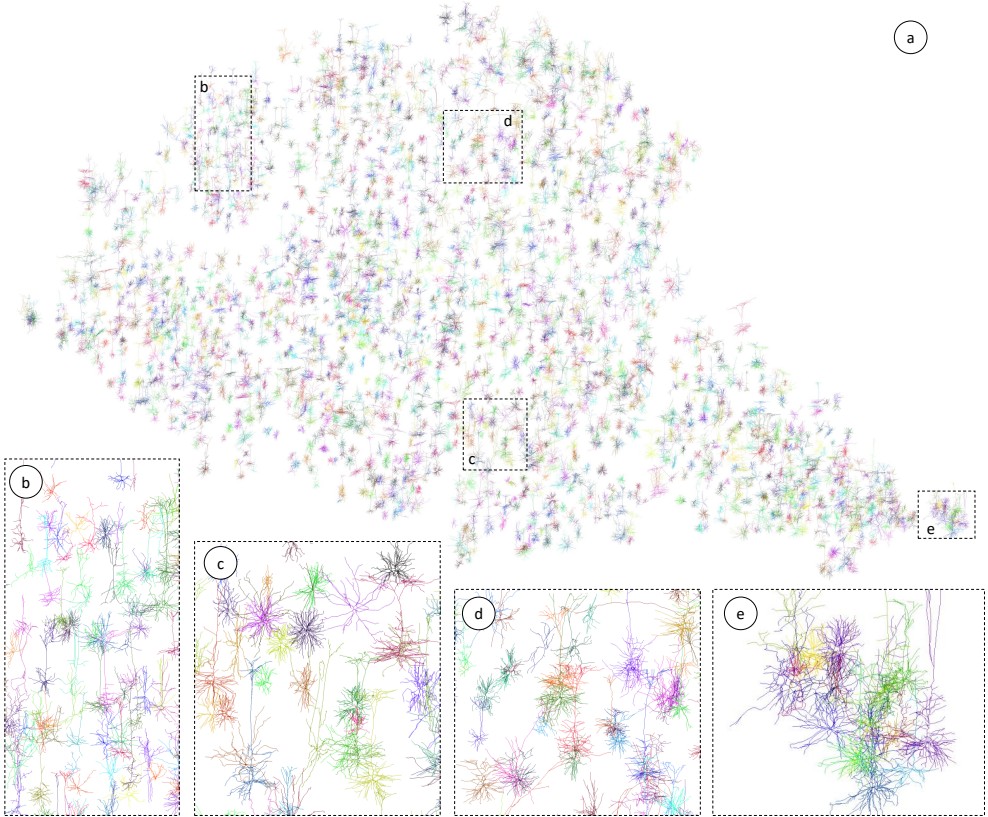

Figure D1: Distribution of neurons in the embedding space of MorphVAE visualized by t-SNE [9]. The thumbnails of dendrites are randomly colored. (b-e) Zoom-in view of regions in (a) showing the mixture of neurons with different morphologies.

## D.2    t-SNE plot of GraphCL

For GraphCL, we can have similar observations as TreeMoCo in Figure 1 — neurons with different morphology can be largely separated in the embedding space. However, as shown in Figure D2(c-e), GraphCL's separation is not as clear as TreeMoCo's. Certain levels of the mixture of neurons with or without apical dendrite can be observed. Since GraphCL relies on local aggregation to generate graph representation, the unique global morphology patterns of apical dendrite could be ignored by it.

## D.3    t-SNE plot of TreeMoCo

Figures D3 to D10 show the high-resolution version figures in Figure 4.

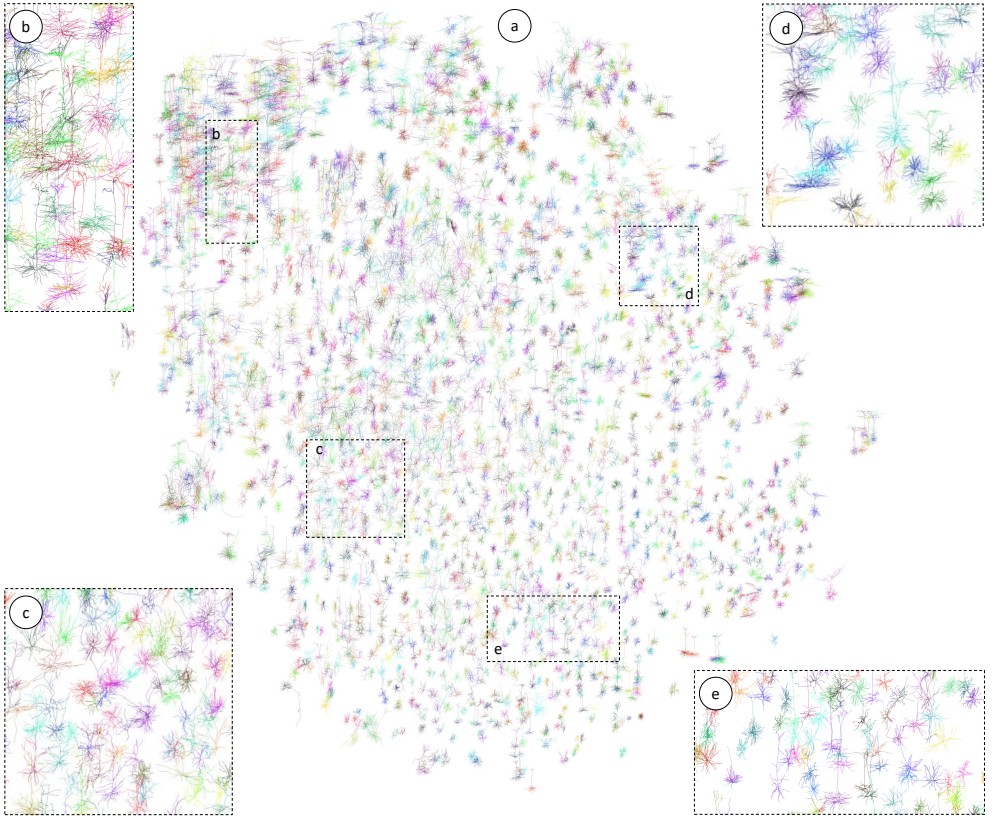

Figure D2: Distribution of neurons in the embedding space of GraphCL visualized by t-SNE [9]. The thumbnails of dendrites are randomly colored. (b-e) Zoom-in view of regions in (a). (c-e) shows the mixture of neurons with and without apical dendrite.

**Algorithm 3** Topology augmentations: Python-like Pseudocodes

```python
# p: applying probability
# max_cnt = 10: the truncation constraint.

cnt = 0 # cnt will be set to 0 at the first call.
def RandomDropSubTrees(self, root):
    reduced_root = Tree(root) # initialize a tree
    if len(root) == 0:
        return reduced_root
    p = np.random.uniform(0, 1, len(root))
    for child_idx in range(len(root)):
        if cnt > max_cnt: # no longger dropping
            reduced_root.append(root[child_idx])
            continue
        else:
            if p[child_idx] > p:
                reduced_root.append(RandomDropSubTrees(root[child_idx])) # recursively traverse
            else:
                cnt += 1 # not appending this child = dropping this child
    return reduced_root

cnt = 0 # cnt will be set to 0 at the first call.
def RandomSwapSiblingSubTrees(root):
    if len(root) < 2:
        return root
    p = np.random.uniform(0, 1, len(root))
    for child_idx in range(len(root)):
        if cnt >= max_cnt:
            break
        if p[child_idx] < p:
            if len(root[child_idx]) < 2:
                continue
            else:
                my_subtree_idx = random.randint(0, len(root[child_idx])-1)
                sibling_idx = random.randint(0, len(root)-1)
                if len(root[sibling_idx]) == 0:
                    continue
                sibling_subtree_idx = random.randint(0, len(root[sibling_idx])-1)
                my_subtree = root[child_idx][my_subtree_idx].copy()
                sibling_subtree = root[sibling_idx][sibling_subtree_idx].copy()
                root[child_idx][my_subtree_idx] = sibling_subtree
                root[sibling_idx][sibling_subtree_idx] = my_subtree
                cnt += 1
        else:
            root[child_idx] = RandomSwapSiblingSubTrees(root[child_idx])
    return root

cnt = 0 # cnt will be set to 0 at the first call.
def RandomSkipParentNode(root):
    if len(root) == 0 or len(root)==1:
        return root
    p = np.random.uniform(0, 1, len(root))
    for child_idx in range(len(root)):
        if cnt >= max_cnt:
            break
        if p[child_idx] < p:
            if len(root[child_idx]) == 0 or len(root[child_idx])==1:
                continue
            else:
                idx = random.randint(0, len(root[child_idx])-1) # picking a child node
                root[child_idx] = root[child_idx][idx] # replacing the parent node with the child node
                cnt += 1
        else:
            root[child_idx] = RandomSkipParentNode(root[child_idx]) # recursively traverse
    return root
```

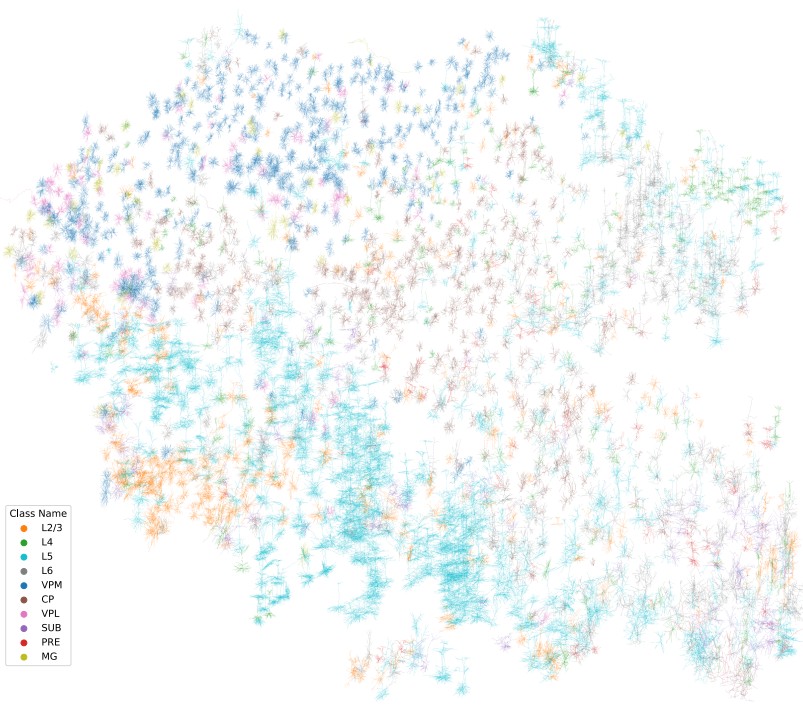

Figure D3: The t-SNE plot of neuron representations colored by class labels. A high-resolution figure for Figure 4-(a).

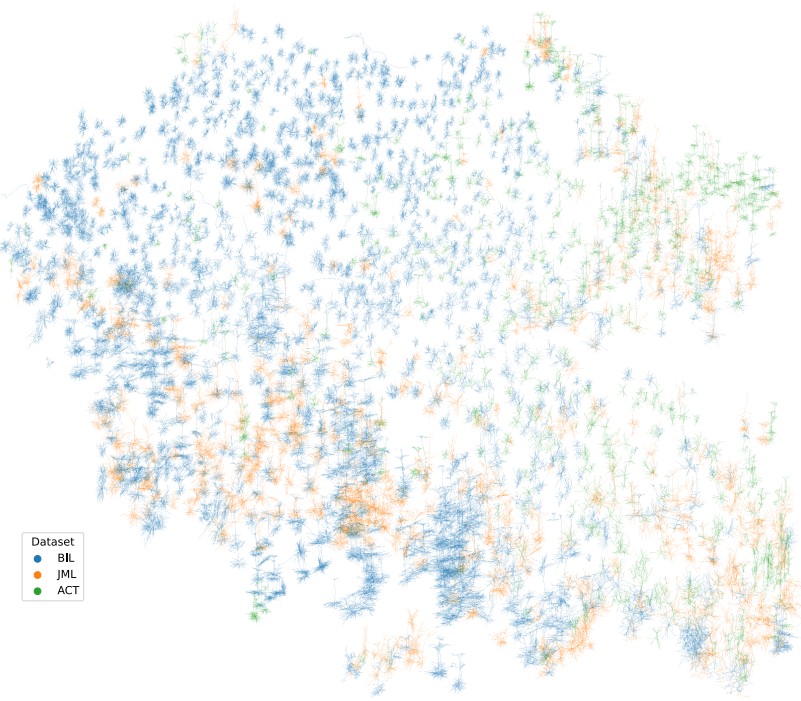

Figure D4: The t-SNE plot of neuron representations colored by dataset sources. A high-resolution figure for Figure 4-(b).

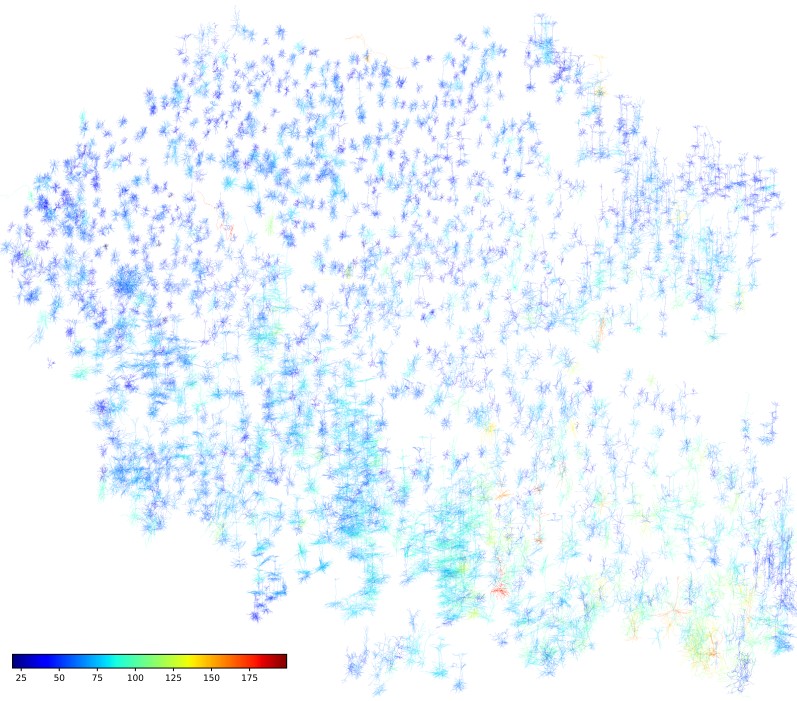

Figure D5: The t-SNE plot of neuron representations colored by the average length of each individual neuron. A high-resolution figure for Figure 4-(c).

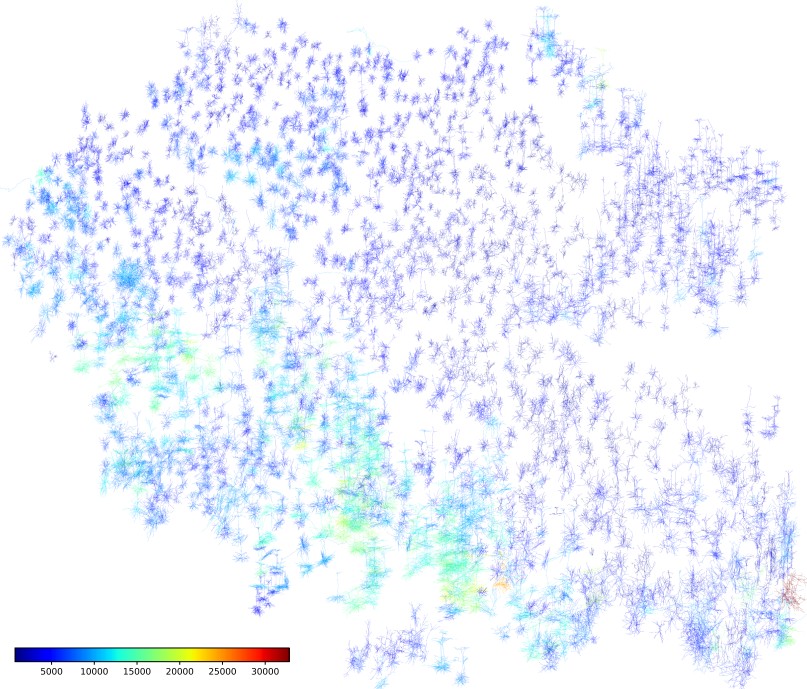

Figure D6: The t-SNE plot of neuron representations colored by the total length of each individual neuron. A high-resolution figure for Figure 4-(d).

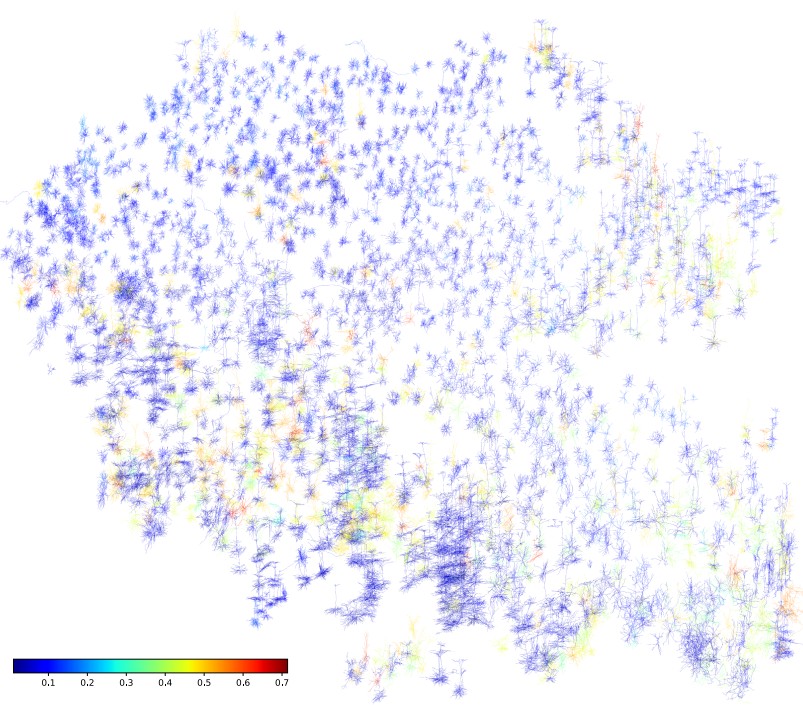

Figure D7: The t-SNE plot of neuron representations colored by the average contraction. A high-resolution figure for Figure 4-(e).

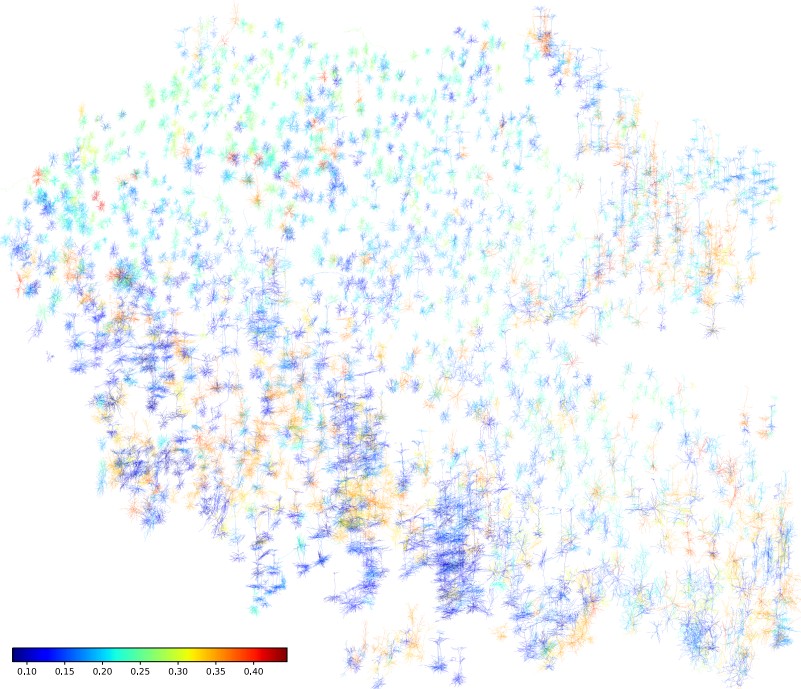

Figure D8: The t-SNE plot of neuron representations colored by the standard deviation of contraction. A high-resolution figure for Figure 4-(f).

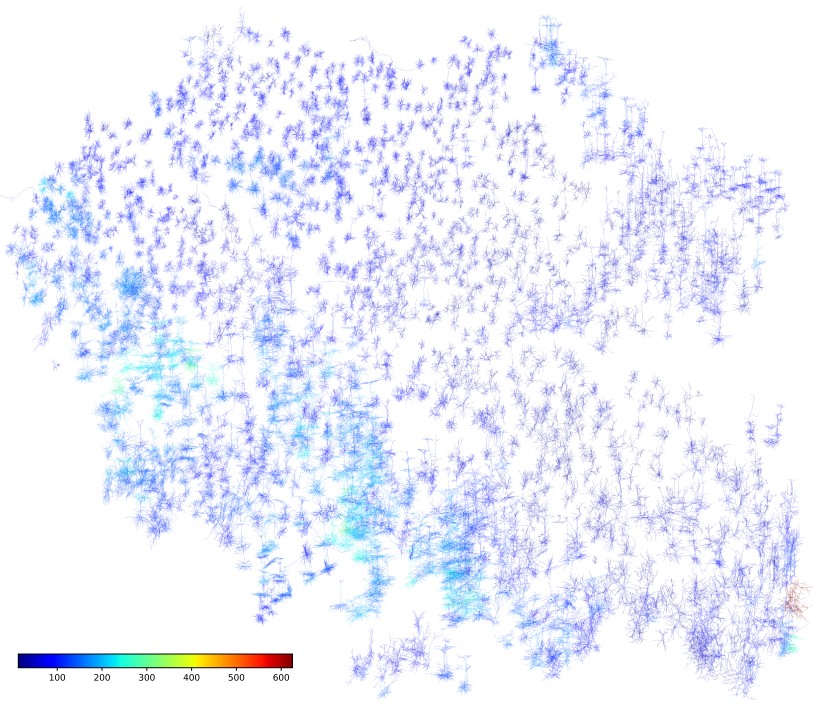

Figure D9: The t-SNE plot of neuron representations colored by the number of nodes inside each individual neuron. A high-resolution figure for Figure 4-(g).

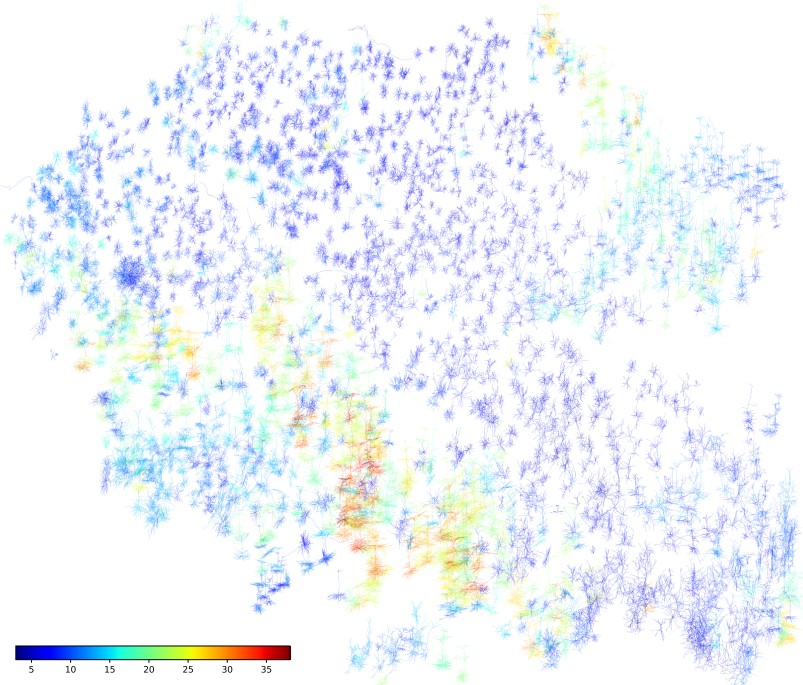

Figure D10: The t-SNE plot of neuron representations colored by the height of each individual neuron tree. A high-resolution figure for Figure 4-(h).