# OpenReview forum: "TreeMoCo: Contrastive Neuron Morphology Representation Learning"
_NeurIPS.cc/2022/Conference — NeurIPS 2022 Accept_

### Official Review · Reviewer_ZR1s · 2022-06-23

**Rating:** 6
**Confidence:** 3
**Soundness:** 2 fair
**Presentation:** 3 good
**Contribution:** 3 good

**Summary:**

The authors represent a neuron as a feature vector for each neuron node, and use an MLP and an Tree-LSTM to embed the neuron vectors in an embedding space using contrastive learning. Augmentations, dedicated to neuron morphology are proposed to enhance the contrastive learning procedure.

**Questions:**

L108: V_i represents a 3D coordinate belonging to vertex i?

L133: What are the input states here? Child nodes?

Eq 1-4: the second subscript of the h-variables was not formally defined. I assume it’s the first and second layer of a stacked LSTM?

L170: are q and k the normalization factors, or the already normalized vectors. Looking at eq. 5 I guess, q and k are the normalized representations, but then the definitions in l170 seem incorrect and should be: q = h(z_q) / ||h(z_q)||_2 etc.

L193: which perturbations are you pointing to here?

L203: What is m?

L301: BN = batch norm, I assume? Why do you find it apparent that this is more needed for self-supervised training? I don’t find this very surprising actually, as there is no label that guides the training, and therefore structure must be sought in the data itself, which might require proper normalization.


**Limitations:**

The training/testing strategy is doubtful (see the quality section above), and the authors did not present any limitations to their proposed work themselves. How is the model, e.g., dependent the type of features that were selected now?

**Strengths And Weaknesses:**

Originality:

I don’t find the method very original or novel. The authors basically use SimCLR and adapt the architecture (inclusion of a Tree-LSTM) and augmentations to be suitable for tree structured data.

Quality:

I am doubting the validity of the performed experiments. There is no separate validation set, and the full training and test set is used to train CoNMoR. That results in two issues. How was the stopping criterium determined for CoNMoR training if no validation set was present? And second, the test set was already seen during CoNMoR pre-training which may highly boost the test set performance when subsequently training a supervised classifier.
Also, in line 264 the authors report the best KNN evaluation accuracy that was acquired during pre-training. So I understand this is not the value after the 100 epochs of pre-training? The authors propose this approach as fully unsupervised, however if labels are used to determine the epoch where KNN accuracy is highest, it is still not fully unsupervised. It would be more fair to report the KNN accuracy at the end of pretraining.
Looking at Fig. 4, it seems that the L5 class is mostly very distinct from the other classes, since almost all metrics show ca clustering pattern around L5. So I’m not too sure whether the representations are really encoding all the features correctly, or simply distinguish L5 from the rest.

Clarity:

Since the paper is aimed for NeurIPS audience, I think it deserves a bit better introduction in the field of neuron morphology and its analyses. Around line 44 I wondered, wat is the input to the model, a 3D reconstruction? But it’s a reconstruction of what? So try to give a better background about these reconstructions, what the data exactly is, how it is measured and what you are reconstructing here. Line 55: what is swc format? Better to explain that as well.
Reading further, the first paragraph of 3.1 is also very unclear to me (I’m a reader without much background in neuron morphology). So if you want to target a broader audience, try to explain the concepts better. Remarks like “Also, the number of such 1-degree
 nodes is biased to neuron reconstruction pipeline and raw data resolution.” need extra explanation.
The last few sentences of this first paragraph in 3.1 might be better understable if a figure was added to illustrate the concept.

Fig 3b is a bit hard to digest, and might need some improvement. Also, Table 4 could have a more descriptive caption. What is the meaning of the numbers and how should the reader compare them to Table 3?

Significance:

The idea of contrastive learning on tree structures is interesting in general. I think, however, that the paper could have had a broader signficance if it was written and tested as a more general method, for any type of tree-structured data. I also wonder whether it is needed to give this method a separate name CoNMoR, since it is basically SimCLR, adapted for tree structured data.

---

> ### Author Response · Authors · 2022-08-02
> **Response to Reviewer ZR1s (1/4)**
>
> We sincerely appreciate the reviewer for careful reading and raising good suggestions to help us improve the paper quality. We have modified the manuscript following the reviewer's suggestions, and below is our response to the reviewer's concerns.
>
> ### 1. Originality
>
> We do adopt a SimCLR-variant --- MoCo (L136) in our work. We are aware that adopting contrastive learning frameworks, e.g., SimCLR or MoCo, for different applications is by no means new, especially when there already exists works introducing contrastive learning frameworks to video, point clouds, and image-text data, and more. Yet, we still want to emphasize our main contribution in tackling neuron tree morphology learning problem by introducing contrastive learning and studying its behavior in this problem --- a new ground for representation learning. With the importance we highlight in the general response, we believe that putting existing machine learning components into a new system to solve a critical neuroscience problem is not trivial and falls under the topics of NeurIPS. Moreover, as pointed out by other reviewers, our method could potentially apply to other applications. This paper could contribute to a new direction for tree representation learning studies. Thus we believe this paper can bring fresh air and fruitful discussion for machine learning research, and we would like to share it with the NeurIPS audience.
>
> We realize that we fail to clearly explain our major contribution in the paper and somehow overclaim our method's novelty. We have revised our manuscript accordingly and renamed CoNMoR to TreeMoCo for a more precise summary of our method.
>
>
>
> ### 2. Qaulity
> **2.1 No validation set, report best epoch on test set, pre-train stop criteria needs label information**
>
> We apologize for not clarifying this point in our initial submission. For all the methods and models we study (supervised and unsupervised), we evaluate them every 5 epochs via a KNN classifier for unsupervised methods or their inherent classification heads for supervised methods and report the best test set performance. Thus, what we report is their best performance during training, following the same protocol (evaluate every 5 epochs) to inspect the best possible performance they can achieve. Though we agree that using a separate validation set is more reasonable, we think our comparisons are also fair, to some extent, to demonstrate the performance difference of different methods since we use the same evaluation protocol for all the methods and results shown in the paper.
>
> Actually, our setting (reporting the best performance in the test set) follows the common setting in self-supervised learning studies, which also aims to demonstrate the best possible performance of different methods. For example, in MoCo's linear evaluation, the fine-tuning model is evaluated every epoch on the test set and the best performance is reported (L292 in the [code](https://github.com/facebookresearch/moco/blob/main/main_lincls.py)). [Wu et al.] use kNN (k=1) to evaluate the performance in the test set during pre-training to select the best model and use K=200 for final evaluation. GraphCL evaluates the models' performance by training a linear SVM classifier on the training set and testing on the test set every 10 epochs (L258 in the [code](https://github.com/Shen-Lab/GraphCL/blob/master/unsupervised_TU/gsimclr.py)). InfoGraph evaluates the models' performance every epoch by training logistics regression, SVM, linear SVM, and random forest classifiers (L128 in the [code](https://github.com/fanyun-sun/InfoGraph/blob/master/unsupervised/main.py)). These evaluation protocols share the same goal --- inspect the best potential of current methods.
>
> In addition, the classification task we use here is just a surrogate to evaluate the quality of the learned embeddings. Our ultimate goal is not to obtain higher accuracy in the classification task but to obtain tree morphology representations that can differentiate neurons with different shapes. One example is to differentiate pyramidal neurons and interneurons, as shown in Figure 1 and discussed in L219-238. These two types of neurons have different shapes and exist in multiple layers of the isocortex - belonging to regions L2/3, L4, L5, and L6 listed in Table 2. In the classification task shown above, they cannot be differentiated as the fine-grained subtype labels are not available. However, with our proposed method, they are clearly separated in the embedding space. Our later analysis has shown that our method also has the potential to unveil new subtypes of neurons and quantify neuronal development stage or deformity - a significant impact on neuroscience research. However, this is beyond the scope of the NeurIPS audience, and thus we will leave these analyses for future works and neuroscience venues.

---

> > ### Author Response · Authors · 2022-08-02
> > **Response to Reviewer ZR1s (2/4)**
> >
> > We recognize that the lack of a stopping criterion is a limitation of our method. Related discussion on this limitation has been added to L307-309:
> > > Moreover, compared to most image contrastive learning frameworks, TreeMoCo's performance on downstream tasks could oscillate after long training epochs on small datasets. An effective early stopping criterion is needed but absent when label information is not available.
> >
> > **2.2 Test set was seen during pre-training**
> >
> > In our initial experiments, the pre-training is conducted on the full dataset, and the downstream task performance is evaluated on certain subsets with the train/test split. We agree that this is confusing and could cause potential data leakage. Our initial idea of pre-training on the full dataset is to answer the question: given a set of neuron trees, can we learn meaningful representation to cluster them? However, as there is no way to quantitatively evaluate the clustering results and compare them with prior supervised arts, we adopt the classification task as a surrogate to evaluate the quality of the learned embeddings. Our ultimate goal is not to obtain higher accuracy in the classification task but to obtain a tree morphology representation that can differentiate neurons with different shapes.
> >
> > To ease the concern, we have replaced the pre-training dataset from the full dataset with the joint training sets (merged training sets of BIL-6, JML-4, and ACT datasets). The results have been updated in Table 3 in the revised submission, which we also summarize below:
> >
> > | Pre-training dataset | BIL-6 | JML-4 | ACT |
> > | -------- | -------- | -------- | -------  |
> > | Full dataset (original) | 79.3 | 62.86 | 54.74 |
> > | Joint training set (revised) | 78.91 | 64.29 | 55.79 |
> >
> > We note that the performance on the JML-4 and ACT datasets is improved, while that of the BIL-6 dataset slightly decreases. We conjecture that this observation is attributed to samples' distribution differences. The full dataset has more classes than the joint training set (See Table 2 for a reference, the grey cells denote classes presented in the joint training set (6 classes), while the full dataset contains all classes except "Others" (10 classes in total).) In addition, the ratio of different data sources accounted for in the pre-training dataset also changes.
> >
> > Overall, new results obtained under a strict train/test split still support our claims. We have replaced all the results obtained from the models pre-trained on the full dataset with the results from the models pre-trained on the joint training set in Table 3.
> >
> > **2.3 Whether the representations are really encoding all the features correctly, or simply distinguish L5 from the rest.**
> >
> > We agree that some classes of neurons are easier to be differentiated from others. However, with some calculations, we can show that our proposed method does not simply distinguish L5 from the rest. The two major classes in BIL-6 data are L5 (315 samples) and VPM (378 samples). The total number of samples is 1279. Assuming all samples from these 2 classes are correctly predicted, the accuracy will be (315+378)/1279=0.54, while the KNN accuracy of our method is around 0.8. The visual inspection by an experienced neuron scientist also confirms the rationality behind the clusters obtained from our method, as discussed in Section 4.2, L219-238 of the revised paper.
> >
> > ### 3. Clarity
> >
> > We thank the reviewer for pointing these issues out. We apologize for the unclear introduction of related neuroscience backgrounds. In our initial submission, we try to include related background information for the NeurIPS audience in section 2. “Background and related works” and section 3.1 “Tree graph representation of neuron”. We have revised them as well as our introduction and figures/tables based on the reviewer’s suggestions. Below we reply to the reviewer’s questions and hope the added information could help the reviewer better assess our work.
> >
> > **3.1 What is 3D neuron reconstruction?**
> >
> > A 3D neuron reconstruction is a tree graph. Each node of the tree has the property of x/y/z coordinate. It is reconstructed from microscope images of brain tissue. It represents the morphology/shape of a neuron tree in brain tissues. An example can be found in Figure 2(a) and more in Appendix A.
> >
> > **3.2 What is swc format?**
> >
> > Swc format is a tabular data format storing neuron tree reconstruction. Each row corresponds to a tree node, and each column corresponds to a property of nodes. We removed the swc part in our revised manuscript as it is a trivial detail for our work.
> >
> > **3.3 Extra explanation of the last few sentences of this first paragraph in 3.1**
> >
> > Based on the reviewer's suggestion, we have rewritten Section 3.1 and adjusted Figure 2 to explain the related concepts better.
> >
> > **3.4 Fig 3b is hard to digest**
> >
> > We have redrawn the figure with fewer lines to make it more clear. We are willing to continue improving it following the reviewer's comming suggestions.

---

> > > ### Author Response · Authors · 2022-08-02
> > > **Response to Reviewer ZR1s (3/4)**
> > >
> > >
> > > **3.5 Table 4, What is the meaning of the numbers and how should the reader compare them to Table 3?**
> > >
> > > We apologize for the insufficient caption of Table 4, which has now been updated in our revised paper. For table 4, as indicated in L271-272, we pre-train on the full BIL-6 dataset (the original submission made a misstatement here, which has been corrected now) for 50 epochs with the augmentations specified by row indices and column indices and report their best KNN accuracy on the BIL-6 test set, following our default evaluation protocol.
> > > For the updated Table 3, we pre-train models with all augmentations for 100 epochs on the joint training sets (merged 3 training sets of BIL-6, JML-4, and ACT datasets).
> > >
> > > The numbers in Tables 3 and 4 cannot be directly compared as the training settings differ. For example, the pre-training datasets and the pre-training epochs are different. In addition, the applying probability of augmentations is different. When applying all the augmentations (Table 3), we use the same probability of 0.2 for them. When ablating augmentations (Table 4), we increase the applying probability to 0.5 to amplify the effect to inspect the augmentations. These settings had been included in our initial Appendix and now are presented in Appendix B.3 of the revised Appendix.
> > >
> > > ### 4. Significance
> > >
> > > We appreciate the reviewer's acknowledgment that: "the idea of contrastive learning on tree structures is interesting in general". We agree that the proposed method has potential as a general method for multiple applications. We have added more discussion to elaborate on this point (L310-315). Meanwhile, we want to convince the reviewer that the problem of learning the representation of neuron morphology is also interesting, challenging, and significant.
> > >
> > > We would like to invite the reviewer to refer to our common reply for a detailed discussion on this and here is a brief summary. Exploration and categorization of neuronal cell-types have been a foundational question in neuroscience studied for over a century and remain an area of intense research focus in public-funded neuroscience. Since dendritic morphology is a key factor of neuronal subtype identity, generating meaningful neuron morphology representation to identify and cluster novel cell-types is of fundamental importance to understand our brain. As we mentioned in our paper, traditional analysis mostly relies on heuristic features and visual inspections to delineate neuron morphology. Only a few machine learning frameworks have been proposed so far to tackle this problem. We hope this work could attract new attention from representation learning and graph/topology society and inspire more future works to solve fundamental needs in neuroscience and unveil our brains' mysteries.
> > >
> > > ###5. Other questions
> > > **5.1 L108: V_i represents a 3D coordinate belonging to vertex i?**
> > >
> > > Yes. V_i = [x_i, y_i, z_i]. We have added it to our revised paper.
> > >
> > > **5.2 L133: What are the input states here? Child nodes?**
> > >
> > > The input states are the hidden states and cell states passed between LSTM units. They are passed from the child nodes to the parent node. We now label the parent and child nodes in Figure 3b for a better illustration. The manuscript has been modified accordingly.
> > >
> > > **5.3 Eq 1-4: are the second subscript of the h-variables the first and second layer of a stacked LSTM?**
> > >
> > > Yes. We have added this information to L112.
> > >
> > > **5.4 L170: are q and k the normalization factors, or the already normalized vectors.**
> > >
> > > Thanks for pointing out the issue! We have corrected it in our revised paper (L150).
> > >
> > > **5.5 L193: which perturbations are you pointing to here?**
> > >
> > > The "perturbations" here are meant to refer to different data augmentations. Augmented samples generated from the same original sample should have similar representations, showing a property of perturbation-invariance. We have replaced it with augmentation for clarity.
> > >
> > > **5.6 L203: What is m?**
> > >
> > > "m" is first shown in L180 (of the revised submission). It is the first letter of "max count", which stands for the truncation threshold. At maximum, "m" subtrees will be augmented, e.g., dropping, skipping, swapping. We have revised the manuscript to make it clearer by referring to "m" with more context:
> > > > L183-184: For simplicity, we empirically use the same m = 10 for (9)-(11) in this paper.

---

> > > > ### Author Response · Authors · 2022-08-02
> > > > **Response to Reviewer ZR1s (4/4)**
> > > >
> > > > **5.7 L301: BN**
> > > >
> > > > Original question: BN = batch norm, I assume? Why do you find it apparent that this is more needed for self-supervised training? I don’t find this very surprising actually, as there is no label that guides the training, and therefore structure must be sought in the data itself, which might require proper normalization.
> > > >
> > > > Response: Yes, BN=batch norm. The statement is based on the observations in Table 5. We agree with the reviewer that it is not a surprising finding to show batch norm is more needed for self-supervised learning. We have revised our discussion around this observation in the manuscript, L286-287:
> > > > > Here, we find that batch normalization (BN) is beneficial for both contrastive self-supervised learning and supervised learning.
> > > >
> > > > ### 6. Limitation
> > > >
> > > > In the revised submission, we have updated Section 5, L303-309 with limitations:
> > > > > We note that TreeMoCo is still a preliminary effort of its kind. Certain limitations can be observed and need to be solved in the future. For instance, Tree-LSTM aggregates the whole graph and thus is sensitive to global changes. This results in its performance decay on incomplete neuron reconstruction (ACT dataset) and its favor of node-feature-based augmentation over topology-based augmentation. Moreover, compared to most image contrastive learning frameworks, TreeMoCo's performance on downstream tasks could oscillate after long training epochs on small datasets. An effective early stopping criterion is needed but absent when label information is not available.

---

> > > > > ### Comment · Reviewer_ZR1s · 2022-08-09
> > > > > **Reply to response**
> > > > >
> > > > > I thank the authors for the extensive rebuttal.
> > > > > In general the clarity of the paper has been improved by adding more background and explanation in some parts of the paper and the authors did take into account the comments that were given.
> > > > >
> > > > > Some question do remain however, with the discussion regarding train/test splits and stopping critera being the most important:
> > > > >
> > > > > 2.1: I see that other related works do use similar evaluation protocols, but even if other researchers take  a certain approach, that does not necesarilly mean it is the best way to do it. In your situation it might for example happen that the test set performance of your model is higher than other models just because there was one training epoch where the performance suddenly peaked (possibly by chance), while the training curve of the benchmarks was much more steady over training. In real-life situations where you want to evaluate your model on new unseen data for which labels are unavailable, you can also not benefit from such a local optimum of your model since you need a general stopping criterium. I think it's good that you added this observation in L308 now, but I would recommend to also emphasize this point more in your results section. To be more transparent, why not reporting the best test set performance for all models, ánd the test set performance after a fixed nr of epochs? Also if you anyway monitor your test set performance throughout training, why not adding the graphs so that the reader can see whether the performance is very fluctuating or stable over time.
> > > > >
> > > > > So to conclude; more analysis/emphasis should be placed on the evaluation protocol in the experiments section, because results can now easily be misinterpreted and models' performance can artifically be boosted now by taking the lucky epoch.
> > > > >
> > > > > In L242: also here explicitly write that you evaluate the best test (as opposed to not mentioning the set) KNN accuracy (similarly as how you mention it for supervised training).
> > > > >
> > > > > 3.3: I appreciate the rewriting of section 3.1, it has definitely improved now. I think the authors forgot to update Figure 2, because I do see a new caption, but subfigures e and f are not visible.
> > > > >
> > > > > 5.7: My question concerning BN was more out of curiosity than that I meant that you should change the meaning of the sentence. Was the observation truly that the unsupervised model benefited more from BN than supervised? In that case i  would still report that, but in the first version you mentioned it as if it was an unexpected finding. Though now the sentence has changed in its meaning suggesting that BN was equally important for both type of models, while it might still be useful information for the reader to know that you observed a difference.
> > > > >
> > > > > I'm inclined to increase my score once a more transparent analysis is added on the stopping criteria and how it influenced the test set performance.

---

> > > > > > ### Author Response · Authors · 2022-08-09
> > > > > > **Response to Reviewer ZR1s**
> > > > > >
> > > > > > We sincerely thank the reviewer for the additional discussions and valuable suggestions. New revised paper and Appendix have been uploaded, with the added text marked in red color.
> > > > > >
> > > > > > **2.1. More analysis/emphasis should be placed on the evaluation protocol in the experiments section.**
> > > > > >
> > > > > > We have included an emphasis on interpreting the results in Section 4.2, L252-256, with a more detailed discussion in Appendix D. We follow the reviewer's suggestions to include the learning curves so that the reader can see whether the performance is very fluctuating or stable over time in Figure C3, page 10 in Appendix. Please refer to the revised paper.
> > > > > >
> > > > > > The added text is summarized below.
> > > > > >
> > > > > > In the main paper:
> > > > > > > It should be emphasized that although the overall test set performance is improved during training, we observe that the test KNN accuracy of both TreeMoCo and GraphCL fluctuates and suffers variations, especially for the smaller JML-4 and ACT datasets. GraphCL fails to improve models over time in the JML-4 dataset under some scenarios. Please refer to Appendix D for more details and discussions, where we provide the performance curves of different methods over training.
> > > > > >
> > > > > >
> > > > > > In Appendix:
> > > > > > > Under our evaluation protocol, all methods' best test set performance is reported. However, the performance evaluated in small datasets like ours usually fluctuates. To get a complete view and the stability of different methods, we provide their performance curves here. Figure C3shows the KNN test set accuracy during training (recorded every 5 epochs). The dashed lines of the same color denote the linear trend lines of different curves. Please refer to Appendix B2-GraphCL and Table B1 for the meaning of "random2/3/4" in Figure C3. The sub-figures' y axes are aligned so that the curves can be directly compared.
> > > > > >
> > > > > > > Despite suffering variance, the overall performance in the BIL-6 dataset is improved over training in all four sub-figures (TreeMoCo and GraphCL). The performance fluctuation of JML-4 and ACT datasets is larger than that of BIL-6. The dataset size matters here since the number of test set samples would affect the granularity of accuracy intervals, exhibiting different degrees of variance. JML-4 and ACT datasets are much smaller than the BIL-6 dataset (about 1.2k/0.3k/0.3k samples for BIL-6, JML-4, and ACT datasets, respectively) and therefore suffer more performance variance. Collecting more data could alleviate this issue. Regarding stability, TreeMoCo and GraphCL perform less stable in the JML-4 and ACT datasets. However, TreeMoCo can still improve the accuracy in the JML-4 dataset over training. In contrast, GraphCL fails to improve that with longer training.
> > > > > >
> > > > > > > With observations from Figure C3, we emphasize the particularity of the neuron tree morphology learning problem and the datasets. Existing large-scale benchmarks in visual or lingual domains usually have abundant training/testing samples and thus can evaluate models precisely with minor variance. However, neuron morphology learning is a new ground to explore. Current insufficient testing samples might make the evaluation noisy and less accurate. We hope this early work could contribute to future exploring in designing a more reliable evaluation protocol.
> > > > > >
> > > > > > We have fixed L242 now.
> > > > > >
> > > > > > **3.3**
> > > > > > The subfigures e and f are in the upper middle of (a) and (b). We will re-emphasize them visually soon (before the rebuttal ends).
> > > > > >
> > > > > > **5.7**: Regarding BN, our statement in the initial submission focuses more on the BIL-6 dataset. The BIL-6 columns in Table 5 (left and right) indicate that: 1) for supervised training, adding BN to the encoder improves it from 83.98% to 84.77%; 2) for self-supervised pre-training, adding BN boosts it from 69.14% to 75.00%, which is a higher gain than that in supervised training.
> > > > > >
> > > > > > However, during our rebuttal period, we found that statement may be misinterpreted as the BN matters much to the ACT dataset (Table 5, left from 50.53% to 63.16%). Unfortunately, we have not ablated BN on the ACT dataset in self-supervised pre-training. Therefore, we modified the statement in the revised paper.
> > > > > >
> > > > > > We initiate our exploration of BN also out of curiosity. Since some self-supervised learning works, e.g., simsiam, and byol, explore the effectiveness of BN in the encoder, projection head, and prediction head. We, therefore, wonder what that case is in our problem.
> > > > > >
> > > > > > Considering the limited time, we will take the reviewer's advice on this part in the camera-ready paper. We could ablate the BN choice in self-supervised learning for JML-4 and ACT datasets, draw a definitive conclusion, and discuss it accordingly.

---

> > > > > > > ### Comment · Reviewer_ZR1s · 2022-08-09
> > > > > > > **Response to reply**
> > > > > > >
> > > > > > > I was unfortunately unable to respond earlier, so I appreciate the effort the authors put in updating the manuscript quickly after my response which was quite late. I think after this full rebuttal both the readability and the transparency on the evaluation protocol have improved.
> > > > > > > I've updated my score accordingly.

---

> ### Author Response · Authors · 2022-08-07
> **Dear reviewer ZR1s: we'd love to know if you have any more questions after our response**
>
> Dear reviewer ZR1s,
>
> We want to thank you again for your helpful feedback and suggestions. They encouraged and helped us to improve the paper. We have tried to carefully address all of your comments in our response and the revised paper. Especially, we conducted more experiments to address concerns about the train/test split and evaluation and elaborated more on neuron morphology and its analyses, and implementation details. Please let us know if you have any further questions, and we are very happy to follow up and keep improving our work!
>
> It matters a lot to us if you found our responses useful and could raise the score! We sincerely appreciate that.
>
> Thank you again for your valuable time and your precious rating!

---

### Official Review · Reviewer_nMVE · 2022-07-09

**Rating:** 6
**Confidence:** 3
**Soundness:** 3 good
**Presentation:** 4 excellent
**Contribution:** 3 good

**Summary:**

This paper presents self-supervised learning for neuron morphology representation. In this paper, the neuron morphology is represented as tree-structured data. Then, the Tree-LSTM is used to represent the tree-structured data. Finally, the representations are learned through contrastive learning with the proposed tree data augmentation techniques.

**Questions:**

In line 203, $m$ is empirically set to 10. What does setting $m$ to 10 mean? Would not changing this value make a difference in performance?

**Ethics Review Area:**

["I don’t know"]

**Limitations:**

Adding some baselines such as graph-based SSL in the quantitative evaluation help show the proposed method's superiority.

**Strengths And Weaknesses:**

The proposed approach for representing neuron morphology is reasonable. Furthermore, this paper is well-written and easy to follow. Some experiments, including ablation studies, are well analyzed and discussed.

However, this paper leaves some negative or uncertain points.
1. First, the proposed framework is too weak from the novelty viewpoint. The proposed representation learning is based on the MoCO [16, 6], one of the visual contrastive learning methods. The differences are that the data is tree-structured and uses tree-data augmentation. Even if the tree data augmentation is novel, some techniques in the augmentation are directly derived from graph data augmentation. As a result, in lines 44 and 307-308, the statement ``this paper proposes a **new SSL framework** for tree structure representation learning'' is an overstatement for the reviewer.

2. Since the tree is a special type of graph, it is necessary to include graph-based SSL [48] as a comparative model in the quantitative evaluation.

3. It would be better to mention/include details of baselines (morpheVAE and TRNN) used in the neuron cell-type classification experiment, such as how to implement and train the baselines. Without the details, it is hard to conclude that the proposed method achieves better performance than baselines.

4. Delineating the learned representations using TSNE was meaningful. However, the reviewer thinks it is not enough to show only the representations of the proposed method. It would be better to include the representations of baselines.

---

> ### Author Response · Authors · 2022-08-02
> **Response to Reviewer nMVE (1/2)**
>
> We thank the reviewer for recognizing the quality of the writings and experiments in this work. Below, we will address the concerns raised by the reviewer.
>
> ### 1. Weak in novelty
> We are aware that we fail to clearly explain our major contribution in the paper and somehow overclaim our method's novelty. We have revised our manuscript accordingly and renamed CoNMoR to TreeMoCo for a more precise summary of our method.
>
> We agree that almost all the major components in this paper have appeared individually in previous works. Rather, our main contribution is to solve the neuron tree morphology learning problem when labels are absent. We believe that putting existing machine learning components into a new system to solve a critical neuroscience problem is not trivial and falls under the topics of NeurIPS. Moreover, as pointed out by other reviewers, our method could potentially apply to other applications of tree-graph representation learning. This paper could contribute to a new direction for tree representation learning studies. Thus we believe this paper can bring new insight and fruitful discussion for machine learning research, and we would like to share it with the NeurIPS audience.
>
>
>
> ### 2. Include graph-based SSL for comparison
>
> We appreciate this constructive comment and follow it to conduct new experiments. We now have compared our method with GraphCL, a graph contrastive learning method, both quantitatively (Table 3 in the revised manuscript) and qualitatively (t-SNE plot in Appendix C.2 in the revised manuscript). Our method performs better than GraphCL on two out of three datasets in KNN evaluation (Table 3) and has better cell-type delineation (Appendix C.2).
>
> In addition, we elaborate on what the fall-behind results of our method could indicate in terms of limitations in L248-251 of the revised submission:
> *GraphCL outperforms TreeMoCo in the ACT dataset but falls behind in BIL-6 and JML-4 datasets, which coincide with the data quality differences (ACT reconstruction is less complete than BIL and JML, more details in Appendix A.2). One explanation is that GraphCL pays more attention to local context and thus is less sensitive to global changes.*
>
> We believe this experiment could contribute to studying the discrepancy between tree graphs and typical constraint-free graphs.
>
> ### 3. Details of baselines
>
> We thank the reviewer for raising this issue! We apologize for not elaborating on this point in our initial submission. We believe all baselines are properly implemented for a fair comparison. More details have been added in revised manuscript L209-216 and Appendix B.2. We briefly summarize them here with additional discussions:
>
> For MorphVAE, we adopt the [officially released code](https://github.com/berenslab/morphvae) and use its default settings to obtain results. Instead of only using the x/y/z coordinate of nodes, we use the same set of processed neurons and 29-d features. The model architecture and the training configurations follow the optimal practice in the original work. The best test set accuracy during 100-epoch training (evaluated every 5 epochs as others) is reported for a fair comparison. TRNN and our method encode the whole tree structure, while MorphVAE encodes the sampled paths inside the trees, which could ignore the tree topology information. And thus, it is not surprising that the performance of TRNN and our TreeMoCo is superior to MorphVAE (Table 3).
>
> For TRNN, the result we provide in the initial submission is from our modified re-implementation (based on the DGL framework). The motivation behind our re-implementation is speed. Our re-implementation can significantly speed up the training process (about ~50X compared to the original code). In our revised submission, we report the results obtained from running the [officially released code](https://github.com/thomasaimondy/treestoolbox/tree/master/casia) under its default settings (which actually have lower performance than our re-implementation in the BIL-6 and JML-4 datasets and higher performance in the ACT dataset). Specifically, in the original code, we change the input layer of the TRNN to take our processed tree graph with 29-d node features as inputs. Similar to evaluations of other methods, we monitor the test set performance every 5 epochs for 100 epochs during training and report the best test accuracy.
>
> In addition to two supervised methods, we now also include a self-supervised graph contrastive learning method, i.e., GraphCL. Please refer to L209-216 and Appendix B.2 in our revised submission for details.

---

> > ### Author Response · Authors · 2022-08-02
> > **Response to Reviewer nMVE (2/2)**
> >
> > ### 4. Include the t-SNE of baselines
> > Following the reviewer's constructive suggestion, we provide the t-SNE plots of MorphVAE and GraphCL in  Appendix C in the revised submission. For MorphVAE, we follow the original work to set the classification loss to zero and only use the reconstruction loss to optimize the model, resulting in a self-supervised training method. For a fair comparison, MorphVAE and GraphCL are unsupervised trained on the same full dataset for 100 epochs (the same setting used by our method). Then, we visualize the learned presentations via t-SNE. The visualization results are shown in Figures 1, C1, and C2 (Figures C1 and C2 are included in the revised Appendix C).
> >
> > For MorphVAE, the visualization shows that MorphVAE largely fails to generate meaningful embeddings to differentiate neuron morphology. This agrees with the observation in the original MorphVAE paper that without a classification loss to train the encoder, MorphVAE's performance on downstream classification tasks would significantly degenerate.
> >
> > While for GraphCL, we can have similar observations as TreeMoCo - neurons with different morphology can be largely separated in the embedding space. However, GraphCL's separation is not as clear as TreeMoCo's. Certain levels of the mixture of neurons with or without apical dendrite can be observed. Since GraphCL relies on local aggregation to generate graph representation, the unique global morphology patterns of apical dendrite could be ignored by it.
> >
> > ### Question: Meaning and selection of "m"
> >
> > "m" is first shown in L180 (of the revised submission). It is the first letter of "max count", which stands for the truncation threshold. At maximum, "m" subtrees will be augmented, e.g., dropping, skipping, swapping. We have revised the manuscript to make it clearer by referring to "m" with more context:
> > > L183-184: For simplicity, we empirically use the same m = 10 for (9)-(11) in this paper.
> >
> > In our experience, changing "m" in a reasonable range (e.g., 5~15) does not affect the performance much.

---

> > > ### Comment · Reviewer_nMVE · 2022-08-10
> > > **Response to Conference Paper6695 Authors**
> > >
> > > I want to thank the authors for the response. Some of questions raised by me are addressed. I raise the score from 5 to 6.

---

> ### Author Response · Authors · 2022-08-07
> **Dear reviewer nMVE: we'd love to know if you have any more questions after our response**
>
> Dear reviewer nMVE,
>
> We sincerely appreciate your recognition of our work and the constructive comments. They encouraged and helped us to improve the paper. We have followed your comments to improve our work, e.g., including a graph-based SSL method, elaborating on baseline implementations and their tSNEs. Please let us know if you have any further questions, and we are very happy to follow up and keep improving our work!
>
> Again, thank you for your valuable time and your precious rating! It means a lot to us if you found our responses address your concerns and further raise the score.

---

### Official Review · Reviewer_DfuA · 2022-07-17

**Rating:** 6
**Confidence:** 4
**Soundness:** 3 good
**Presentation:** 3 good
**Contribution:** 3 good

**Summary:**

This paper proposed a contrastive framework to learn features from neuron trees. The authors have devised a carefully engineered set of augmentation suitable for tree-structured data sets. The learned features were used to classify cell types that yield superior performance compared to the previous methods.

**Questions:**

See the comments on weakness.

**Limitations:**

The authors have not adequately commented on the known limitations.

**Strengths And Weaknesses:**

Strengths:
1. The proposed method is technically sound and novel.
2. The authors proposed several augmentations for the neuron trees, which turned out to be useful for self-supervised feature learning.
3. The results are convincing across multiple datasets showing generalizability.

Weaknesses:
1. While the proposed method yields superior performance in particular problems for neuron morphology learning, my concern is that it is a niche problem. How could it benefit the general graph/topology learning community in general? The authors should comment on the applicability of the proposed tree representation learning in other domains, for example, molecular properties classification.
2. The authors did not elaborate on the motivation for using InfoNCE loss over several other contrastive settings. Is there any particular reason to choose it?
3. The authors mentioned a few contrastive learning on graphs. However, a direct comparison is missing to demonstrate the advantage over the previous approaches.
4. It is unclear whether the baseline experiments [21,51] are performed in identical experimental settings. Further, if one has also to include the axons, would the tree structure assumption still hold? What is ‘m’ mentioned in the data augmentation?

---

> ### Author Response · Authors · 2022-08-02
> **Response to Reviewer DfuA (1/2)**
>
> We sincerely thank the reviewer for the careful and constructive comments. We are glad for and encouraged by your acknowledgment of our work's technical novelty and soundness. Below is our response to the weakness of this work in the reviewer's comments.
>
>
> ### 1. How could it benefit the general graph/topology learning community?
>
> Thanks for this suggestion! We agree with the reviewer that the proposed method can be adopted for other tree morphology/topology studies, including molecular properties prediction. Following the reviewer's suggestion, more discussion has been added in the revised manuscript L310-315.
>
> Meanwhile, we would like to argue that learning the morphology representation of neuron trees is a significant task by itself. Exploration and categorization of neuronal cell-types have been a foundational question in neuroscience studied for over a century and remain an area of intense research focus in public-funded neuroscience. Since dendritic morphology is a key factor of neuronal subtype identity, generating meaningful neuron morphology representation to identify and cluster novel cell-types is of fundamental importance to understand our brain. As we mentioned in our paper, traditional analysis mostly relies on heuristic features and visual inspections to delineate neuron morphology. Only a few machine learning frameworks have been proposed so far to tackle this problem. We hope this work could attract new attention from representation learning and graph/topology society and inspire more future works to solve fundamental needs in neuroscience and unveil our brains' mysteries. We would like to invite the reviewer refering to our common reply for more details.
>
> ### 2. Motivation for using InfoNCE loss
>
> Chen et al. explored three choices of contrastive losses, i.e., InfoNCE loss, a Margin Triplet loss, and a normalized temperature-scaled logistic loss, and found InfoNCE loss to be the best in their SimCLR paper. We thus follow them to use InfoNCE loss. Exploring different options will be a good idea, which we leave for future work. We now have explained this choice in the revised manuscript L140-141.
>
> Reference:
> Chen, Ting, et al. "A simple framework for contrastive learning of visual representations." In ICML 2020.
>
>
> ### 3. Comparison to graph contrastive learning methods
>
> We appreciate this constructive comment and follow it to conduct new experiments. We now have compared our method with GraphCL, a graph contrastive learning method, both quantitatively (Table 3 in the revised manuscript) and qualitatively (t-SNE plot in Appendix C.2 in the revised manuscript). Our method performs better than GraphCL on two out of three datasets in KNN evaluation (Table 3) and has better cell-type delineation (Appendix C.2).
>
> In addition, we elaborate on what the fall-behind results of our method could indicate in terms of limitations in L248-251 of the revised submission:
> *GraphCL outperforms TreeMoCo in the ACT dataset but falls behind in BIL-6 and JML-4 datasets, which coincide with the data quality differences (ACT reconstruction is less complete than BIL and JML, more details in Appendix A.2). One explanation is that GraphCL pays more attention to local context and thus is less sensitive to global changes.*
>
> We believe this experiment could contribute to studying the discrepancy between tree graphs and typical constraint-free graphs.

---

> > ### Author Response · Authors · 2022-08-02
> > **Response to Reviewer DfuA (2/2)**
> >
> >
> > ### 4. Experiments setting
> >
> > **4.1 Baseline implementation**
> >
> > We thank the reviewer for raising this issue! We apologize for not elaborating on this point in our initial submission. We believe all baselines are properly implemented for a fair comparison. More details have been added in revised manuscript L209-216 and Appendix B.2. We briefly summarize them here with additional discussions:
> >
> > For MorphVAE, we adopt the [officially released code](https://github.com/berenslab/morphvae) and use its default settings to obtain results. Instead of only using the x/y/z coordinate of nodes, we use the same set of processed neurons and 29-d features. The model architecture and the training configurations follow the optimal practice in the original work. The best test set accuracy during 100-epoch training (evaluated every 5 epochs as others) is reported for a fair comparison. TRNN and our method encode the whole tree structure, while MorphVAE encodes the sampled paths inside the trees, which could ignore the tree topology information. And thus, it is not surprising that the performance of TRNN and our TreeMoCo is superior to MorphVAE (Table 3).
> >
> > For TRNN, the result we provide in the initial submission is from our modified re-implementation (based on the DGL framework). The motivation behind our re-implementation is speed. Our re-implementation can significantly speed up the training process (about ~50X compared to the original code). In our revised submission, we report the results obtained from running the [officially released code](https://github.com/thomasaimondy/treestoolbox/tree/master/casia) under its default settings (which actually have lower performance than our re-implementation in the BIL-6 and JML-4 datasets and higher performance in the ACT dataset). Specifically, in the original code, we change the input layer of the TRNN to take our processed tree graph with 29-d node features as inputs. Similar to evaluations of other methods, we monitor the test set performance every 5 epochs for 100 epochs during training and report the best test accuracy.
> >
> > In addition to two supervised methods, we now also include a self-supervised graph contrastive learning method, i.e., GraphCL. Please refer to L209-216 and Appendix B.2 in our revised submission for details.
> >
> > **4.2 If one has also to include the axons, would the tree structure assumption still hold?**
> >
> > Yes, the architecture of axons is also a tree rooted at neuron soma (same as dendrite). Our proposed method can be directly adopted to generate the representation of neuron axons. However, fusing the representation of axon and dendrite will be tricky as an axon is typically much larger than a dendrite. Also, due to its large size, the cost of generating axon reconstruction is significantly higher than that of a dendrite, which limits the amount of axon reconstruction that can be analyzed. Thus, in this study, we only focus on neuron dendrite for simplicity and applicability reasons. We would like to continue exploring our method's application to neuron axons in the future.
> >
> >
> > **4.3 What is ‘m’ mentioned in the data augmentation?**
> >
> > "m" is first shown in L180 (of the revised submission). It is the first letter of "max count", which stands for the truncation threshold. At maximum, "m" subtrees will be augmented, e.g., dropping, skipping, swapping. We have revised the manuscript to make it clearer by referring to "m" with more context:
> >
> > >  L183-184:  For simplicity, we empirically use the same m = 10 for (9)-(11) in this paper.
> >
> > **4.4 Limitations**
> >
> > In the revised submission, we have updated Section 5, L303-309 with limitations:
> > > We note that TreeMoCo is still a preliminary effort of its kind. Certain limitations can be observed and need to be solved in the future. For instance, Tree-LSTM aggregates the whole graph and thus is sensitive to global changes. This results in its performance decay on incomplete neuron reconstruction (ACT dataset) and its favor of node-feature-based augmentation over topology-based augmentation. Moreover, compared to most image contrastive learning frameworks, TreeMoCo's performance on downstream tasks could oscillate after long training epochs on small datasets. An effective early stopping criterion is needed but absent when label information is not available.

---

> ### Author Response · Authors · 2022-08-07
> **Dear reviewer DfuA: we'd love to know if you have any more questions after our response**
>
> Dear reviewer DfuA,
>
> We sincerely appreciate your acknowledgment of our work and the helpful suggestions. They encouraged and helped us to improve the paper. We have tried to carefully address all of your comments in our response and the revised paper. Please let us know if you have any further questions, and we are very happy to follow up and keep improving our work!
>
> Thank you again for your valuable time and your precious rating!

---

> ### Comment · Reviewer_DfuA · 2022-08-08
> **Renponse to Conference Paper6695 Authors**
>
> I thank the authors for the revised submission. The added details improved the paper. Hence I stick to my original rating.

---

### Official Review · Reviewer_esRK · 2022-07-18

**Rating:** 6
**Confidence:** 3
**Soundness:** 2 fair
**Presentation:** 3 good
**Contribution:** 3 good

**Summary:**

This paper proposes to apply contrastive self-supervised representation learning to tree-structured neuron morphology data. The authors adopted Tree LSTM (Ref. [51]) and MOCO (Ref. [16]), and proposed tree topology data augmentation. The proposed method was demonstrated on three public datasets (BIL, JML, and ACT). Learned representation was visualized with 2D t-SNE and then qualitatively inspected and analyzed. For quantitative evaluation, cell-type classification accuracy was measured with (1) K Nearest-Neighbor (KNN) accuracy from self-supervised pre-training and (2) supervised fine-tuning.

**Questions:**

* (L224) Was the training/test set split only used in supervised fine-tuning? Why not in pre-training?
* (L235-236) The choice of number of neighbors for KNN is very arbitrary.
* (L264) "..., with the best KNN evaluation accuracy during pre-training included."
    * I do not understand what it exactly means. Do you record KNN evaluation accuracy when each epoch is done, and report the best among them? That doesn't seem very reasonable to me. I think if you were to report cell-type classification accuracy from pre-training, you should strictly use separate training and test sets, and report KNN evaluation accuracy on the test set **after** pre-training is done.
* (L287-288) Why pre-training for 100 epochs for the main experiment and 50 epochs for the ablation study?

**Limitations:**

See "Weaknesses" in the above.

**Strengths And Weaknesses:**

### Strengths
This paper is written clearly and presented well. The problem of interest (representation learning on unlabeled neuron morphology data) is clearly significant.


### Weaknesses
Novelty is limited. This paper has two novelties: (1) application of contrastive self-supervised learning to tree-structured neuron morphology data, and (2) topology data augmentation for tree data. However, the ablation study (Table 4 and Section 4.3) shows that the proposed topology data augmentation may not be effective nor important. Table 4 shows that the point transformation ("point") is the single-most effectective one, and the combination of "point" and attribute masking ("mask") achieved the best accuracy. The "point" and "mask" are both general graph augmentation, so the claimed novelty of the proposed tree topology augmentation becomes rather pointless.

Evaluation contains multiple problems. First, I don't understand why the Allen Cell Types (ACT) dataset was included in the evaluation. As is shown in Figure A2, ACT's neuron reconstruction is largely incomplete. I see no point in using incomplete reconstructions which could only complicate representation learning. Second, training and evaluation seem quite arbitrary. Why was the training/test set split only used in supervised fine-tuning, but not in pre-training? It is very weird to compute KNN accuracy **during, not after, pre-training and report the best one** while not even having a separate test set. Also why pre-training for 100 epochs for the main experiment and 50 epochs for the ablation study? Third, it is unclear whether the two baselines (MorphVAE and TRNN) were faithfully reproduced. Without any evidence for this, the main quantitative result of this paper in Table 3 cannot be fully justified.

---

> ### Author Response · Authors · 2022-08-01
> **Response to Reviewer esRK (1/3)**
>
>
> We thank the reviewer for acknowledging the quality and significance of our work. The comments are addressed below.
>
> ### 1. Novelty
>
> We are aware that almost all the major components in this paper have appeared individually in previous works. One new component in our framework is the proposed customized augmentation. Though the reviewer pointed out that tree topology augmentation is not as effective as point augmentation methods, the point augmentations we designed are also customized for neuron morphology data. We will discuss this later. It should be noted that our main contribution is to solve the neuron tree morphology learning problem when labels are absent. We believe that putting existing machine learning components into a new system to solve a critical neuroscience problem is not trivial and falls under the topics of NeurIPS. Moreover, as pointed out by other reviewers, our method could potentially apply to more applications related to the tree structure, such as program source code representation learning. This paper could contribute to a new direction for tree representation learning studies. Thus we believe this paper can bring new insight and fruitful discussion for machine learning research, and we would like to share it with the NeurIPS audience.
>
> To avoid overclaiming the novelty in methodology, we rename CoNMoR to TreeMoCo and revise our claims in the manuscript accordingly.
>
>
> #### Tree topology augmentation is less effective than point feature augmentation
>
> We thank the reviewer for raising this point. It is interesting that tree topology augmentation does not generate good results as point-feature-based augmentation. This finding is quite intriguing for us as the augmentation is designed following tree characteristics. This could be caused by the characteristics of neuron reconstruction and Tree-LSTM, as discussed below. We believe it is still worth showing this result as it can bring some useful discussion and guidance for future work.
>
>
> More discussions on possible causes have been added in L277-281:
> *Topology augmentations do not necessarily bring better performance. This could be caused by the mechanism of Tree-LSTM, which aggregates the whole tree to generate embeddings and thus is sensitive to the global changes caused by topology augmentation. Better neuron morphology representations might be excepted if a more proposer set of augmentations or a better network architecture were applied under the TreeMoCo framework.*
>
>
> ### 2. Evaluation contains multiple problems
> **2.1 ACT data is incomplete**
>
> Classifying neurons with incomplete morphology reconstruction is like detecting an object with occlusion – challenging but meaningful. We agree with the reviewer that this will complicate the representation learning problem. However, this type of data with incomplete morphology is typical in neuroscience studies due to the limitation of some microscopes and the high cost of generating complete reconstruction. Thus, to show if and to which degree our proposed method could work on incomplete reconstruction is essential for the application. Notably, this dataset actually is not in favor of our proposed method and shows the potential limitations of our method in dealing with incomplete neuron trees. More discussion has been added to the new manuscript L304-306 and Appendix L33-36.

---

> > ### Author Response · Authors · 2022-08-02
> > **Response to Reviewer esRK (2/3)**
> >
> > **2.2 Train/test split problem**
> >
> > In our initial experiments, the pre-training is conducted on the full dataset, and the downstream task performance is evaluated on certain subsets with the train/test split. We agree that this is confusing and could cause potential data leakage. Our initial idea of pre-training on the full dataset is to answer the question: given a set of neuron trees, can we learn meaningful representation to cluster them? However, as there is no way to quantitatively evaluate the clustering results and compare them with prior supervised arts, we adopt the classification task as a surrogate to evaluate the quality of the learned embeddings. Our ultimate goal is not to obtain higher accuracy in the classification task but to obtain a tree morphology representation that can differentiate neurons with different shapes.
> >
> >
> > To ease the concern, we have replaced the pre-training dataset from the full dataset with the joint training sets (merged training sets of BIL-6, JML-4, and ACT datasets). The results have been updated in Table 3 in the revised submission, which we also summarize below:
> >
> > | Pre-training dataset | BIL-6 | JML-4 | ACT |
> > | -------- | -------- | -------- | -------  |
> > | Full dataset (original) | 79.3 | 62.86 | 54.74 |
> > | Joint training set (revised) | 78.91 | 64.29 | 55.79 |
> >
> > We note that the performance on the JML-4 and ACT datasets is improved, while that of the BIL-6 dataset slightly decreases. We conjecture that this observation is attributed to samples' distribution differences. The full dataset has more classes than the joint training set (See Table 2 for a reference, the grey cells denote classes presented in the joint training set (6 classes), while the full dataset contains all classes except "Others" (10 classes in total).) In addition, the ratio of different data sources accounted for in the pre-training dataset also changes.
> >
> > Overall, new results obtained under a strict train/test split still support our claims. We have replaced all the results obtained from the models pre-trained on the full dataset with the results from the models pre-trained on the joint training set in Table 3.
> >
> > **2.3 Training epochs difference between main experiments and ablation studies**
> >
> > The setting difference between the pre-training and ablation training epoch is meant to save computation budgets. This resembles settings in different works, e.g., MoCo pre-trains for 200 epochs for the main experiments and 100 epochs for the ablation study, and SimCLR pre-trains for 800 epochs for the main experiments and 100 epochs for the ablation study.
> >
> > **2.4 Whether the two baselines were faithfully reproduced**
> >
> > We thank the reviewer for raising this issue! We apologize for not elaborating on this point in our initial submission. We believe all baselines are properly implemented for a fair comparison. More details have been added in revised manuscript L209-216 and Appendix B.2. We briefly summarize them here with additional discussions:
> >
> > For MorphVAE, we adopt the [officially released code](https://github.com/berenslab/morphvae) and use its default settings to obtain results. Instead of only using the x/y/z coordinate of nodes, we use the same set of processed neurons and 29-d features. The model architecture and the training configurations follow the optimal practice in the original work. The best test set accuracy during 100-epoch training (evaluated every 5 epochs as others) is reported for a fair comparison. TRNN and our method encode the whole tree structure, while MorphVAE encodes the sampled paths inside the trees, which could ignore the tree topology information. And thus, it is not surprising that the performance of TRNN and our TreeMoCo is superior to MorphVAE (Table 3).
> >
> > For TRNN, the result we provide in the initial submission is from our modified re-implementation (based on the DGL framework). The motivation behind our re-implementation is speed. Our re-implementation can significantly speed up the training process (about ~50X compared to the original code). In our revised submission, we report the results obtained from running the [officially released code](https://github.com/thomasaimondy/treestoolbox/tree/master/casia) under its default settings (which actually have lower performance than our re-implementation in the BIL-6 and JML-4 datasets and higher performance in the ACT dataset). Specifically, in the original code, we change the input layer of the TRNN to take our processed tree graph with 29-d node features as inputs. Similar to evaluations of other methods, we monitor the test set performance every 5 epochs for 100 epochs during training and report the best test accuracy.
> >
> > In addition to two supervised methods, we now also include a self-supervised graph contrastive learning method, i.e., GraphCL. Please refer to L209-216 and Appendix B.2 in our revised submission for details.

---

> > > ### Author Response · Authors · 2022-08-02
> > > **Response to Reviewer esRK (3/3)**
> > >
> > >
> > > ## 3. Other questions
> > >
> > > **3.1 The choice of number of neighbors for KNN is very arbitrary**
> > >
> > > Since the size of neuron morphology datasets is relatively small, we use [MoCo's settings](https://colab.research.google.com/github/facebookresearch/moco/blob/colab-notebook/colab/moco_cifar10_demo.ipynb) on the Cifar10 dataset as a major reference. The Cifar10 dataset has 6000 images per class, and the number of neighbors for KNN is 200. In addition, in [Wu et al.], they set the number of neighbors for KNN also as 200 for imagenet, which has 1200 images per class. We change that number from 200 to 20, simply diving 10 to match the scale of our dataset. In our experience, changing the number of neighbors for KNN does not affect the results too much.
> > >
> > > **3.2 Report metrics of best epoch**
> > >
> > > We apologize for not clarifying this point in our initial submission. During training, we evaluate models every 5 epochs via a KNN classifier for unsupervised methods or their inherent classification heads for supervised methods and report the best test set performance for all the studied methods. Thus, all the results reported in the paper are the best test performance they achieved during our experiments. This setting (reporting the best performance in the test set) is common in self-supervised learning studies and also demonstrates the best possible performance of different methods. For example, in MoCo's linear evaluation, the fine-tuning model is evaluated every epoch on the test set, and the best performance is reported (L292 in the [code](https://github.com/facebookresearch/moco/blob/main/main_lincls.py)). [Wu et al.] use kNN (k=1) to evaluate the performance in the test set during pre-training to select the best model and use K=200 for final evaluation. GraphCL evaluates the models' performance by training a linear SVM classifier on the training set and testing on the test set every 10 epochs (L258 in the [code](https://github.com/Shen-Lab/GraphCL/blob/master/unsupervised_TU/gsimclr.py)). InfoGraph evaluates the models' performance every epoch by training logistics regression, SVM, linear SVM, and random forest classifiers (L128 in the [code](https://github.com/fanyun-sun/InfoGraph/blob/master/unsupervised/main.py)).
> > >
> > >
> > >
> > > **Reference**:
> > > Wu, Zhirong, et al. "Unsupervised feature learning via non-parametric instance discrimination." In CVPR 2018.
> > >
> > > You, Yuning, et al. "Graph contrastive learning with augmentations." In NeurIPS 2020.
> > >
> > > Sun, Fan-Yun, et al. "Infograph: Unsupervised and semi-supervised graph-level representation learning via mutual information maximization." In ICLR 2020.

---

> ### Author Response · Authors · 2022-08-07
> **Dear reviewer esRK: we'd love to know if you have any more questions after our response**
>
> Dear reviewer esRK:
>
> We want to thank you again for your helpful suggestions and the problems raised in the comments. They helped us to improve the paper. We have tried to carefully address all of your comments in our response and the revised paper. Especially, we elaborated on the novelty issue, and conducted more experiments to address concerns about the train/test split and evaluation. Please let us know if you have any further questions, and we are very happy to follow up and keep improving our work!
>
> It means a lot to us if you found our responses convincing and could raise the score! We sincerely appreciate that.
>
> Thank you for your valuable time and your precious rating!

---

> > ### Comment · Reviewer_esRK · 2022-08-09
> > **After rebuttal**
> >
> > Dear Authors,
> >
> > Thank you for addressing the issues I raised. Most of the issues are well addressed to at least an "understandable" level. However, reporting the best performance during training cannot be justfied by saying "this setting (reporting the best performance in the test set) is common in self-supervised learning studies." I personally view this as a bad tradition that has to be eliminated. SSL has to be used in practice to be meaningful and useful, but such a tradition only confuses practitioners who have to decide when to stop. Does the epoch with the best test set performance guarantee the best possible learned representation and the best downstream performance? It's hard to know. Therefore I decided to increase my score to 5: Borderline accept.

---

> > > ### Comment · Reviewer_ZR1s · 2022-08-09
> > > **Reply to reviewer esRK**
> > >
> > > I totally agree with the remark of reviewer esRK: just because other researchers got through reviewing phases while having questionable practices does not mean that we as a research community should blindly follow it without being critical.
> > >
> > > The authors did, however, add the test accuracy graphs over epochs in Appendix D now, which does add valuable insights and transparency. But looking at the graphs and seeing that for mainly only one of the three datasets the difference in the proposed methods vs benchmarks seems significant, I raised my score not higher than a 6.

---

> > > > ### Author Response · Authors · 2022-08-09
> > > > **Response to reviewers esRK and ZR1s**
> > > >
> > > > We thank reviewers esRK and ZR1s for the time in improving our work and the discussion raised here.
> > > >
> > > > We cannot agree more with both reviewers in questioning current practices in some SSL works. The problem (when to stop) has been bothering many researchers, including us, for a long while. This issue emerges almost everywhere in practice for different SSL studies and also exists for other hyperparameter selections. For example, our experience with other established SSL methods shows that switching the linear evaluation training configs between some SSL works (e.g., using MoCo's settings for BYOL or SimCLR evaluation) can bring a 2% to 8% accuracy drop in the ImageNet dataset. The training epoch, batch size, learning rate, weight decay, optimizer settings, and the pre-trained models jointly contribute to the final linear evaluation results.
> > > >
> > > > It will be interesting and essential to study how to select the best hyper-parameters without label information for SSL frameworks. Solving this problem alone will be a significant work. However, we do not have a good answer for it now, and we think it is slightly beyond the scope of our current work. Thus, as discussed in the limitation section, we will keep searching for a possible solution and leave it for our future work. In the new ground of neuron morphology learning problem, our current work uses the same evaluation protocol for all the methods, which we think is relatively fair but not the best way.
> > > >
> > > > Again, we sincerely appreciate the reviewers' efforts and time spent on our work. They helped us to improve our work a lot and leave valuable discussions on current evaluation protocols within and beyond the context of this work.

---

> > > > > ### Comment · Reviewer_esRK · 2022-08-10
> > > > > **Score update**
> > > > >
> > > > > I thank reviewer ZR1s for pointing out that the test accuracy graphs over epochs has been added to Appendix D. I missed it before and now agree that it's quite informative, and I do appreciate the transparency it adds. I decided to increase my score to 6: Weak accept.

---

### Author Response · Authors · 2022-08-02
**General Response**


We thank reviewers for their positive and constructive comments. We have renamed our method as TreeMoCo, following reviewers' suggestions. Detailed responses to reviewers' comments will be addressed in the follow-ups of each reviewer's comments. In addition, we have revised our paper and supplemental Appendix accordingly, with major changes marked by blue color. If not otherwise specified, the line numbers we use during the first-round rebuttal refer to the line numbers in the revised paper.

Here, we would like to argue that learning the morphology representation of neuron trees is a significant task by itself. Exploration and categorization of neuronal cell-types have been a foundational question in neuroscience studied for over a century (Waldeyer, 1891) and remain an area of intense research focus in public-funded neuroscience. For example, two-thirds of topics in $5 billion BRAIN 2.0 NIH fundings (Ngai, 2022) are closely related to neuronal cell-type analysis. Another example is NeuroMorpho.Org - a database of digital neuron reconstructions that has averaged over 34,000 site visits and 178,000 downloads per month for 15 years (Ascoli et al., 2017). Lacking attention from machine learning society does not indicate the problem itself is niche.

A neuron is the basic computation unit composing brain intelligence. However, a consensus on the definition of neuron cell-types is still absent. Since dendritic morphology is a key factor of neuronal subtype identity, generating meaningful neuron morphology representation to identify and cluster novel cell-types is of fundamental importance to address these project needs. Moreover, computing such representation is the first step in establishing the quantitative correlation between neuron morphology and other neuron properties such as gene expression or connectivity. The present work fulfills a critical need of this aim by probing the composition of extremely dense brain structures made up of tens of thousands of individual neurons.

One fact of the data used in this paper is that they are very new and of extremely high scientific value. The JML dataset was published in *Cell* in 2019, and the BIL dataset was published in *Nature* in 2021. The availability of such data creates new opportunities and needs for representation learning. We want to share this information with the machine learning society together with our initial efforts.

As we mentioned in our paper, traditional analysis mostly relies on heuristic features and visual inspections to delineate neuron morphology. Only a few machine learning frameworks have been proposed so far to tackle this problem. Furthermore, to our best knowledge, this is one of the earliest attempts to ease the need for label information with self-supervised tree morphology representation learning. We hope this work could attract new attention from representation learning and graph/topology society and inspire more future works to solve fundamental needs in neuroscience and unveil our brains' mysteries.

References:

Waldeyer, W, von (1891): Uber einige neuere Forshungen im Gebiete der Anatomie des centralen Nervensystems. Deutsch. Med. Wochenschr.17: 1213-1218; 1244-1246; 1267-1269; 1331-1332; 1352-1356.

Ngai J. BRAIN 2.0: Transforming neuroscience. Cell. 2022 Jan 6;185(1):4-8. doi: 10.1016/j.cell.2021.11.037. PMID: 34995517.

Ascoli GA, Maraver P, Nanda S, Polavaram S, Armañanzas R. Win-win data sharing in neuroscience. Nat Methods. 2017 Jan 31;14(2):112-116. doi: 10.1038/nmeth.4152. PMID: 28139675; PMCID: PMC5503136.

---

### Meta-Review · Area_Chair_VbCv · 2022-09-07

**Recommendation:** Accept
**Confidence:** Certain

**Metareview:**

The paper proposes a self-supervised learning method targeting representation learning for large-scale neuronal morphology. This is an important problem in connectomics, and systems neuroscience at large, and very little progress has been to date in applying modern machine learning methods to this problem. The authors combine a TreeLSTM with self-supervised learning.

The paper received borderline reviews, but the reviewers were unanimous in recommending acceptance. Two main concerns were raised. First, the reviewers questioned the novelty and applicability of the method for the machine learning field at large. Ultimately, there was some acknowledgement that some of the techniques proposed could have applicability beyond the field of ML for neuronal morphology. But even if this were not the case, neuroscience is an important focus area for the NeurIPS conference. Second, there were issues raised with the evaluation methodology. These were clarified by the authors, and additional material was added to the Appendix to enable a better understanding of the method.

I recommend acceptance.

**Award:**

No

---

### Decision · Program_Chairs · 2022-09-14

Accept